# Towards Fully Parameter-Free Stochastic Optimization: Grid Search with Self-Bounding Analysis

**Yuheng Zhao** [1 2]  **Yu-Hu Yan** [1 2]  **Amit Attia** [3]  **Tomer Koren** [3]  **Lijun Zhang** [1 2]  **Peng Zhao** [1 2]

## Abstract

Parameter-free stochastic optimization aims to design algorithms that are agnostic to the underlying problem parameters while still achieving convergence rates competitive with optimally tuned methods. While some parameter-free methods do not require the specific values of the problem parameters, they still rely on prior knowledge, such as the lower or upper bounds of them. We refer to such methods as "partially parameter-free". In this work, we target achieving "*fully* parameter-free" methods, i.e., the algorithmic inputs do not need to satisfy any *unverifiable* condition related to the true problem parameters. We propose a powerful and general *grid search* framework, named GRASP, with a novel *self-bounding* analysis technique that effectively determines the search ranges of parameters, in contrast to previous work. Our method demonstrates generality in: *(i)* the non-convex case, where we propose a fully parameter-free method that achieves near-optimal convergence rate, up to logarithmic factors; *(ii)* the convex case, where our parameter-free methods are competitive with strong performance in terms of acceleration and universality. Finally, we contribute a sharper guarantee for the model ensemble, a final step of the grid search framework, under interpolated variance characterization.

## 1. Introduction

Stochastic optimization is a fundamental problem in machine learning and optimization (Bottou et al., 2018; Lan, 2020), with applications across various domains, such as the training of large-scale models (Kingma & Ba, 2015; Loshchilov & Hutter, 2019), reinforcement learning (Schulman et al., 2017; Shao et al., 2024), and so on.

One of the most fundamental stochastic optimization methods is Stochastic Gradient Descent (SGD) (Robbins & Monro, 1951), which uses stochastic gradients estimated from mini-batch samples for the update. The success of SGD depends largely on the choice of step size. In practice, a common approach is to define lower and upper bounds for the step size and perform a grid search within this range to find the optimal tuning. Typically, heuristic bounds such as $10^{-5}$ and $10^1$ are used, which lack theoretical guarantees.

Meanwhile, optimization theory indicates the optimal step size depends on problem parameters: in the non-convex case, on the initial function suboptimality gap and objective properties (e.g., smoothness, noise bound) (Ghadimi & Lan, 2013); in the convex case, on the Lipschitz constant and the initial distance to the optimum (Nemirovski et al., 2009). Since these parameters are often unavailable in practice, research has increasingly focused on *parameter-free* stochastic optimization (Li & Orabona, 2019; Ward et al., 2019; Mcmahan & Streeter, 2012; McMahan & Orabona, 2014; Carmon & Hinder, 2022), where non-convex and convex methods are developed without specific values of the problem parameters. A detailed review of related works is provided in Appendix A. However, despite aiming for parameter-freeness, most works still require inputs tied to certain problem parameters, e.g., the state-of-the-art method in stochastic convex optimization, U-DoG (Kreisler et al., 2024), requires a coarse upper bound on gradient noise and a lower bound on the initial distance to the optimum.

We introduce a more rigorous characterization of parameter-freeness to distinguish between "*partially*" or "*fully*" parameter-free methods, enabling clearer comparisons across existing approaches. Specifically, an algorithm is considered fully parameter-free with respect to a problem parameter $X$ if it is not only agnostic to the specific value of $X$ but also does not rely on any *unverifiable* condition related to $X$ (e.g. upper or lower bounds). Otherwise, it is classified as *partially* parameter-free.

We aim to develop fully parameter-free methods by adopting the *grid search* framework, a common approach in optimiza-

[1]State Key Laboratory for Novel Software Technology, Nanjing University, China [2]School of Artificial Intelligence, Nanjing University, China [3]Tel Aviv University, Israel. Correspondence to: Peng Zhao <zhaop@lamda.nju.edu.cn>.

*Proceedings of the $43^{rd}$ International Conference on Machine Learning*, Seoul, South Korea. PMLR 306, 2026. Copyright 2026 by the author(s).

*Table 1.* Comparison of parameter-freeness and convergence rates for stochastic optimization. Notations: $\bar{X} \triangleq \max\{X, X_\varepsilon\}$ for $X \in \{L_\ell, F_\ell, L_\nu, d_0\}$ with *any* $X_\varepsilon > 0$; '✗', '✔', and '✓' denote no, partial, and full parameter-freeness; $\delta$ is the confidence level.

*(a)* **Non-convex case.** $L_\ell$ is smoothness parameter, $F_\ell \triangleq \ell(\mathbf{x}^0) - \ell(\mathbf{x}^\star)$, $\Delta_\ell$ is maximum gradient noise, $\widehat{\mathbf{g}}^0$ is empirical value of $\nabla\ell(\mathbf{x}^0)$. The result of Attia & Koren (2024) is partially free because the lower and upper bounds of the learning rate, i.e., $\eta_{\min}$ and $\eta_{\max}$, require prior knowledge of the problem parameters.

| Reference | Freeness | | | Convergence Rate of $\|\nabla\ell(\mathbf{x}^{\mathrm{out}})\|$ |
|---|---|---|---|---|
| | $L_\ell$ | $F_\ell$ | $\Delta_\ell$ | |
| Tuned SGD (Ghadimi & Lan, 2013) | ✗ | ✗ | ✗ | $\sqrt{\frac{L_\ell F_\ell \Delta_\ell^2}{T}} + \frac{L_\ell F_\ell + \Delta_\ell^2(\log\frac{1}{\delta})}{T}$ |
| Attia & Koren (2024, Theorem 1) | ✔ | ✔ | ✔ | $\sqrt{\frac{L_\ell F_\ell \Delta_\ell^2}{T}\left(\log_+ \frac{\eta_{\max}}{\eta_{\min}}\right)\left(\log\frac{1}{\delta}\right)} + \frac{L_\ell F_\ell + \Delta_\ell^2(\log\frac{1}{\delta})}{T}\left(\log_+ \frac{\eta_{\max}}{\eta_{\min}}\right)\left(\log\frac{1}{\delta}\right)$ |
| **Ours [Theorem 2]** | ✓ | ✓ | ✓ | $\sqrt{\frac{\bar{L}_\ell \bar{F}_\ell \Delta_\ell^2}{T}\left(\log_+ \frac{T\|\widehat{\mathbf{g}}^0\|}{L_\varepsilon F_\varepsilon}\right)} + \frac{\bar{L}_\ell \bar{F}_\ell + \Delta_\ell^2(\log\frac{1}{\delta})^2}{T}\left(\log_+ \frac{T\|\widehat{\mathbf{g}}^0\|}{L_\varepsilon F_\varepsilon}\right)$ |

*(b)* **Convex case.** $L_\ell$ and $(L_\nu, \nu)$ are (Hölder) smoothness parameters, $G$ is Lipschitz constant, $d_0 \triangleq \|\mathbf{x}^0 - \mathbf{x}^\star\|$, $\Delta(\cdot)$ is gradient noise function, $\Delta_D$ is maximum noise defined in Eq. (2). $\sigma_\star$ is value noise at $\mathbf{x}^\star$. $\widehat{\mathbf{g}}^0$ and $\widehat{\ell}^0$ are empirical values of $\nabla\ell(\mathbf{x}^0)$ and $\ell(\mathbf{x}^0)$. The † marks adaptivity to Hölder smoothness. Meanings of colors: main convergence, variance, optimum's noise, and ensemble error.

| Reference | Freeness | | Requirements (Conditions) | Convergence Rate of $\ell(\mathbf{x}^{\mathrm{out}}) - \ell(\mathbf{x}^\star)$ |
|---|---|---|---|---|
| | $L_\ell$ | $d_0$ | | |
| Kavis et al. (2019) | ✓† | ✗ | Domain bounded by $D$ | $\frac{L_\nu D^{1+\nu}}{T^{\frac{1+3\nu}{2}}} + \frac{D\Delta_D}{\sqrt{T}}\left(\log\frac{1}{\delta}\right)$ |
| Kreisler et al. (2024) | ✓ | ✗ | $\widehat{\Delta}(\cdot)$ ; $r_\varepsilon$ 
 $(\geq\Delta(\cdot))$ $(\leq d_0)$ | $\left(\min\left\{\frac{L_\ell d_0^2}{T^2}, \frac{Gd_0}{\sqrt{T}}\right\} + \frac{d_0\Delta_{2d_0}}{\sqrt{T}} + \frac{d_0\widehat{\Delta}_{2d_0}}{T}\right)\left(\log_+ \frac{d_0}{r_\varepsilon}\right)^2\left(\log\frac{1}{\delta}\right)^2\left(\log_+ T\frac{\widehat{\Delta}_{2d_0}+\min\{L_\ell d_0^2, Gd_0\}}{\ell(\mathbf{x}_0)-\ell(\mathbf{x}^\star)}\right)^4$ |
| **Ours [Theorem 3]** | ✓ | ✓ | — | $\frac{L_\ell d_0^2}{T^2}\left(\log_+ \frac{\bar{d}_0}{d_\varepsilon}\right)^2 + \frac{\bar{d}_0\Delta_{2\bar{d}_0}}{\sqrt{T}}\left(\log_+ \frac{\bar{d}_0}{d_\varepsilon}\right)^{\frac{1}{2}}\left(\log\frac{1}{\delta}\right) + \mathrm{ERR}\left(T / \left(\log_+ \frac{T\|\widehat{\mathbf{g}}^0\|}{L_\varepsilon d_\varepsilon}\right)\right)$ |
| **Ours [Theorem 4]** | ✓† | ✓ | $\ell_\varepsilon^\star$ 
 $(\leq\ell(\mathbf{x}^\star))$ | $\frac{\bar{L}_\nu \bar{d}_0^{1+\nu}}{T^{\frac{1+3\nu}{2}}}\left(\log_+ \frac{\bar{d}_0}{d_\varepsilon}\right)^{\frac{1+3\nu}{2}} + \frac{\bar{d}_0\Delta_{2\bar{d}_0}}{\sqrt{T}}\left(\log_+ \frac{\bar{d}_0}{d_\varepsilon}\right)^{\frac{1}{2}}\left(\log\frac{1}{\delta}\right) + \sigma_\star\sqrt{\frac{\log(1/\delta)}{T}} + \mathrm{ERR}\left(T / \left(\log_+ \frac{T(\widehat{\ell}^0-\ell_\varepsilon^\star)}{L_\varepsilon d_\varepsilon}\right)\right)$ |

tion practice. Existing grid search methods are partially parameter-free (Attia & Koren, 2024; Khaled & Jin, 2024), as they rely on *given* ranges of problem parameters to determine the search range. However, in real-world scenarios, the ranges of problem parameters, especially their upper bounds, are often unavailable.

To this end, we propose a general and powerful grid search framework, along with a novel "*self-bounding*" analysis technique to address the lack of theoretical guidance in determining the search ranges of problem parameters. Our technique effectively derives computable search ranges for unknown problem parameters, while ensuring full parameter-freeness and maintaining competitive theoretical guarantees. Our methodology has been proven effective for both non-convex and convex optimization.

Our results significantly enrich the line of research on parameter-free stochastic optimization for both non-convex and convex cases, offering a series of new contributions:

- For fully parameter-free stochastic optimization, we propose a methodology that enhances the general grid search framework with a novel *self-bounding* analysis technique. This technique effectively derives the parameter search ranges without any prior knowledge of the problem parameters, as described in Section 3.1.

- For non-convex and smooth stochastic optimization, we propose a *fully* parameter-free method w.r.t. *all* problem parameters, with convergence rate matching the optimal one up to logarithmic factors, as stated in Table 1a.

- For convex and smooth stochastic optimization, we propose a *fully* parameter-free method w.r.t. *all* problem parameters, producing a candidate with near-optimal *accelerated* convergence up to logarithmic factors. For the more general Hölder smoothness, our method demonstrates *fully* parameter-freeness w.r.t. both the Hölder smoothness parameters and the distance to the optimum, producing a candidate with *universal* convergence, competitive with the state-of-the-art method of Kreisler et al. (2024). These results are summarized in Table 1b.

- We provide a *new* guarantee for model ensemble with an interpolated variance characterization, thereby enhancing grid search, as summarized in Table 2.

**Organization.** The rest of the paper is organized as follows. In Section 2, we introduce the preliminaries. In Section 3, we provide a general introduction to our grid search methodology. In Section 4 and Section 5, we provide the theoretical results for non-convex and convex optimization, respectively. Then in Section 6, we conduct experiments to verify the effectiveness of the proposed methods. Finally, we conclude the paper in Section 7. Due to page limits, most proofs are deferred to the appendix.

## 2. Preliminaries

In this section, we introduce optimization problem setups and definitions of parameter-freeness used in this paper.

**Notations.** We use $[N]$ to denote the index set $\{1, \ldots, N\}$. The $\ell_2$-norm is denoted by $\|\cdot\|$. Independently sampling from the oracle $\mathcal{Q}(\cdot)$ at $\mathbf{x}$ for $t$ times is denoted by $\mathcal{Q}_1(\mathbf{x}), \ldots, \mathcal{Q}_t(\mathbf{x})$. We define $\log_+(x) \triangleq 1 + \log(x)$. We omit the $\log \log$ factor in the asymptotic notation $\mathcal{O}(\cdot)$.

### 2.1. Optimization Problem Setups

We focus on unconstrained stochastic optimization:

$$\min_{\mathbf{x} \in \mathbb{R}^d} \ell(\mathbf{x}), \tag{1}$$

where we denote the optimum by $\mathbf{x}^\star \in \arg\min_{\mathbf{x} \in \mathbb{R}^d} \ell(\mathbf{x})$. The algorithm starts from the initial point $\mathbf{x}^0$ and outputs a solution $\mathbf{x}^{\text{out}}$. During the optimization process, we assume access to two stochastic oracles: one for gradients and one for function values, which is common in the machine learning and stochastic optimization literature.

**Assumption 1.** There exists a first-order oracle $\mathbf{g}(\cdot)$ such that: given any $\mathbf{x} \in \mathbb{R}^d$, it satisfies $\mathbb{E}[\mathbf{g}(\mathbf{x})] = \nabla \ell(\mathbf{x})$ and $\Pr[\|\mathbf{g}(\mathbf{x}) - \nabla \ell(\mathbf{x})\| \le \Delta(\mathbf{x}) \le \Delta_\ell] = 1$, where $\Delta : \mathbb{R}^d \to \mathbb{R}_+$ is an *unknown* function and $\Delta_\ell$ is an *unknown* constant.

**Assumption 2.** There exists a zeroth-order oracle $\widetilde{\ell}(\cdot)$ such that: given any $\mathbf{x} \in \mathbb{R}^d$, it satisfies $\mathbb{E}[\widetilde{\ell}(\mathbf{x})] = \ell(\mathbf{x})$ and $\Pr[|\widetilde{\ell}(\mathbf{x}) - \ell(\mathbf{x})| \le \sigma_\ell] = 1$, with an *unknown* constant $\sigma_\ell$.

Specifically, we interpret sampling from the oracle as drawing a differentiable function $\ell_\xi(\cdot)$ parameterized by the random variable $\xi$, where $\widetilde{\ell}(\cdot) = \ell_\xi(\cdot)$ and $\mathbf{g}(\cdot) = \nabla \ell_\xi(\cdot)$ represent the zeroth- and first-order oracles, respectively. This notation is omitted when the context makes it clear.

We then define the following maximum noise bound for a domain with center $\mathbf{x}^0$ and diameter $D$ as:

$$\Delta_D \triangleq \max_{\|\mathbf{x} - \mathbf{x}^0\| \le D} \Delta(\mathbf{x}). \tag{2}$$

In the following, we provide the formal definitions of smoothness, Lipschitz continuity, and Hölder smoothness.

**Definition 1** (Smoothness). The objective $\ell(\cdot)$ is $L_\ell$-smooth if $\|\nabla \ell(\mathbf{x}) - \nabla \ell(\mathbf{y})\| \le L_\ell \|\mathbf{x} - \mathbf{y}\|$ for all $\mathbf{x}, \mathbf{y} \in \mathbb{R}^d$.

**Definition 2** (Lipschitz Continuity). The objective $\ell(\cdot)$ is $G$-Lipschitz if $|\ell(\mathbf{x}) - \ell(\mathbf{y})| \le G\|\mathbf{x} - \mathbf{y}\|$ for all $\mathbf{x}, \mathbf{y} \in \mathbb{R}^d$.

**Definition 3** (Hölder Smoothness). The objective $\ell(\cdot)$ is $(L_\nu, \nu)$-Hölder smooth with $L_\nu > 0, \nu \in [0, 1]$, if $\|\nabla \ell(\mathbf{x}) - \nabla \ell(\mathbf{y})\| \le L_\nu \|\mathbf{x} - \mathbf{y}\|^\nu$ for all $\mathbf{x}, \mathbf{y} \in \mathbb{R}^d$.

We consider two setups: non-convex and convex objective.

For non-convex optimization, the objective $\ell(\cdot)$ is not necessarily convex but is $L_\ell$-smooth. We focus on the high-probability convergence rate of the squared gradient norm, $\|\nabla \ell(\mathbf{x}^{\text{out}})\|^2$, and the optimal rate of the tuned SGD relies on the functional gap between the initial point and the optimum, i.e. $F_\ell \triangleq \ell(\mathbf{x}^0) - \ell(\mathbf{x}^\star)$, the maximum gradient noise $\Delta_\ell$, and the smoothness parameter $L_\ell$.

For convex optimization, we focus on the high-probability convergence rate of the sub-optimality gap $\ell(\mathbf{x}^{\text{out}}) - \ell(\mathbf{x}^\star)$. For accelerated convergence, we assume the objective is $L_\ell$-smooth. For the more challenging universal convergence, we assume that the objective function satisfies one of the following cases: *(i)* either $L_\ell$-smooth or $G$-Lipschitz; *(ii)* $(L_\nu, \nu)$-Hölder smooth. The optimal rate relies on the distance between the initial point and the optimum $d_0 \triangleq \|\mathbf{x}^0 - \mathbf{x}^\star\|$, the maximum gradient noise $\Delta_\ell$, and the smoothness parameters $L_\ell$ or $(L_\nu, \nu)$.

### 2.2. Definition of Parameter-Freeness

In this part, we introduce partial and full parameter-freeness in order to better distinguish between different levels of parameter dependence and to facilitate a clear comparison with previous works. First, we assume the algorithm has access to the oracle query budget $T$ and the confidence level $\delta$. Regarding the requirement for prior knowledge of problem parameters, such as $\Delta_\ell, \sigma_\ell, L_\ell, F_\ell, d_0$ defined in Section 2.1, we propose the following definitions.

**Definition 4** (Partial / Full Parameter-Freeness). An optimization method is *parameter-free* w.r.t. some problem parameter $X$ if it does not require $X$ as an input. Formally,

- **Partial parameter-freeness:** It is agnostic to $X$, but some algorithmic inputs must satisfy an *unverifiable* condition related to $X$. For example, the input $X'$ must satisfy that $X' \le X$, but there is no guarantee that this condition holds.

- **Full parameter-freeness:** Its algorithmic inputs do not need to satisfy any *unverifiable* conditions related to $X$.

It is important to note that parameter-free does not imply freeness from *all* parameters; rather, it specifically refers to the absence of key problem parameters related to the problem's properties, such as $\Delta_\ell, \sigma_\ell, L_\ell, F_\ell, d_0$ defined in Section 2.1. In the rest of the paper, we abbreviate the above definitions as "partially / fully free to $X$" for convenience.

## 3. Our Grid Search Framework

In this section, we introduce our grid search framework, GRASP (GRid-seArch with Self-bounding for Parameter-free Optimization), including three key components: self-bounding analysis, budget allocation, and model ensemble.

**Framework Overview.** Given an algorithm $\mathcal{A}$ with parameters to be tuned, such as the step size, and a total oracle budget $T$, the grid search framework proceeds as follows:

(1) Identify a suitable range for each algorithmic parameter, where our self-bounding technique assists in doing this.

(2) Discretize the search range and allocate the budget; for each value in the discretized set, run an instance of the algorithm $\mathcal{A}$ using the corresponding parameter values.

(3) Select the best output from these algorithm runs.

Then, in Section 3.2, we use deterministic convex and smooth optimization as an example to illustrate the entire pipeline of the grid search methodology.

### 3.1. General Idea of Self-Bounding Analysis

The grid search framework is commonly used in optimization practice; however, it typically relies on manually specified ranges for algorithmic inputs, lacking theoretical guidance for selecting these ranges. To address this, we propose the "self-bounding" analysis technique as a simple yet effective solution. We use convex and smooth optimization as an illustrative example and then present the general idea.

**An Intuitive Example.** Consider the convex and smooth optimization with *deterministic gradients*, where the optimal convergence is $\mathcal{O}(L_\ell d_0^2/T^2)$, with $d_0$ being the initial distance to the optimum. Since $d_0$ could be infinitely large, it is difficult to determine a suitable search range for it. Our key insight is that, *this $d_0$ is naturally bounded to ensure the accelerated rate is non-vacuous*. Specifically, the target rate has a *quadratic* dependence on $d_0$, so if $d_0$ is excessively large, the convergence rate will be worse than a trivial bound, e.g., $\ell(\mathbf{x}^0) - \ell(\mathbf{x}^\star) \leq \|\nabla\ell(\mathbf{x}^0)\| d_0$. To this end, we use this trivial bound as a benchmark to derive a reasonable upper bound for $d_0$, that is, by solving the inequality:

$$\|\nabla\ell(\mathbf{x}^0)\| d_0 \geq \frac{\max\{L_\ell, L_\varepsilon\} d_0^2}{T^2} \geq \frac{L_\ell d_0^2}{T^2}, \qquad (3)$$

thereby leading to a computable and effective upper bound of $d_0 \leq \|\nabla\ell(\mathbf{x}^0)\| T^2/L_\varepsilon$ with arbitrary $L_\varepsilon > 0$.

This example illustrates the main idea of our self-bounding analysis: we can determine an *effective* search range for unknown problem parameters, given appropriate connections between the target and benchmark rates. Below, we present a more rigorous formulation of this idea.

**Self-Bounding Analysis.** Consider the target rate of:

$$\epsilon^{\text{TAR}}\left(T, p^1, \ldots, p^m\right), \qquad (4)$$

where $p^1, \ldots, p^m \in \mathbb{R}_+$ are unknown problem parameters, and $\epsilon^{\text{TAR}}$ is a non-decreasing function of these parameters. For convex and smooth optimization, the target rate is $\epsilon^{\text{TAR}}(T, L_\ell, d_0) = L_\ell d_0^2/T^2$, which depends on the smoothness parameter $L_\ell$ and the initial distance $d_0$.

For each parameter $p^i$, we require a *user-specified* input $p^i_\varepsilon > 0$, which can generally be very small. Next, we define

$\bar{p}^i \triangleq \max\{p^i, p^i_\varepsilon\}$, and shift our target to:

$$\epsilon^{\text{TAR}}\left(T, \bar{p}^1, \ldots, \bar{p}^m\right) \geq \text{Eq. (4)}. \qquad (5)$$

In our convex and smooth optimization example, our target rate becomes $\mathcal{O}(\bar{L}_\ell \bar{d}_0^2/T^2)$, where $\bar{L}_\ell \triangleq \max\{L_\ell, L_\varepsilon\}$ and $\bar{d}_0 \triangleq \max\{d_0, d_\varepsilon\}$. Note that the user-specified $L_\varepsilon$ and $d_\varepsilon$ do *not* necessarily satisfy any conditions w.r.t. $L_\ell$ or $d_0$, which does not conflict with the full parameter-freeness.

Subsequently, we find a point $\mathbf{x}^{\text{BM}}$ with a trivial benchmark convergence rate of $\epsilon^{\text{BM}}$. In our example, simply choosing $\mathbf{x}^{\text{BM}} = \mathbf{x}^0$ yields $\epsilon^{\text{BM}} \triangleq \|\nabla\ell(\mathbf{x}^0)\| d_0$.

Next we can derive an upper bound for grid search. For an unknown parameter $p^i$, solving the following inequality:

$$p^i_{\max} = \arg\max_{p^i}\left\{\epsilon^{\text{BM}} \geq \epsilon^{\text{TAR}}(T, \bar{p}^1, \ldots, p^i, \ldots, \bar{p}^m)\right\}$$

and we set $p^i_{\max}$ as an effective *upper bound* for $p^i$. This is not the theoretically largest upper bound, but if $p^i$ exceeded it, the target rate would become worse than that of the benchmark $\epsilon^{\text{BM}}$, making the final rate vacuous. Therefore, to achieve the target rate in Eq. (5), it suffices to focus on the range of $[p^i_\varepsilon, \max\{p^i_\varepsilon, p^i_{\max}\}]$ for each $\bar{p}^i$.

Finally, we ensemble all candidates within the grids, including the benchmark $\mathbf{x}^{\text{BM}}$, as in Attia & Koren (2024); Khaled & Jin (2024), to ensure the following convergence:

$$\epsilon^{\text{TAR}}\left(T, \bar{p}^1, \ldots, \bar{p}^m\right) \text{poly} \log\left(\frac{p^1_{\max}}{p^1_\varepsilon}, \ldots, \frac{p^m_{\max}}{p^m_\varepsilon}\right). \quad (6)$$

In general, our result (6) will eventually suffer a logarithmic factor on $1/p^i_\varepsilon$ for each $i \in [m]$. Thus, setting a small user-specified $p^i_\varepsilon$ will *not* significantly affect the final rate.

### 3.2. An Initialization for Deterministic Optimization

In this part, we illustrate our framework within the deterministic convex smooth optimization setting as a clearer demonstration. As shown above, for the target rate of $\mathcal{O}(\bar{L}_\ell \bar{d}_0^2/T^2)$, the self-bounding analysis provides a search range of $[d_\varepsilon, d_{\max}]$ for $\bar{d}_0$, where $d_{\max} \triangleq \max\{d_\varepsilon, \|\nabla\ell(\mathbf{x}^0)\| T^2/L_\varepsilon\}$ with arbitrary $L_\varepsilon, d_\varepsilon > 0$.

**Discretization and Allocation.** We discretize the range $[d_\varepsilon, d_{\max}]$ using a geometric sequence, i.e., let $D_i = d_\varepsilon 2^i$ for $i \in [N]$, where $N = \lceil \log_2(d_{\max}/d_\varepsilon) \rceil$. We allocate an oracle budget of $T_i$ for each UNIXGRAD (Kavis et al., 2019), a base algorithm for convex smooth optimization, fed with $D_i$. In the case where there exists $i_\star \in [N]$ such that $D_{i_\star}/2 \leq \bar{d}_0 \leq D_{i_\star}$, the $i_\star$-th UNIXGRAD ensures:

$$\mathcal{O}\left(\frac{\bar{L}_\ell D_{i_\star}^2}{T_{i_\star}^2}\right) \leq \mathcal{O}\left(\frac{\bar{L}_\ell \bar{d}_0^2}{T_{i_\star}^2}\right). \qquad (7)$$

Then a simple *uniform* allocation of $T_i = \lfloor \frac{T}{N} \rfloor$ yields:

$$\mathcal{O}\left(\frac{\bar{L}_\ell \bar{d}_0^2}{T_{i_\star}^2}\right) = \mathcal{O}\left(\frac{\bar{L}_\ell \bar{d}_0^2}{T^2}\left(\log_+ \frac{d_{\max}}{d_\varepsilon}\right)^2\right). \qquad (8)$$

**Algorithm 1** Deterministic Convex and Smooth OPT

**Input:** Oracle budget $T$, initial point $\mathbf{x}^0$, $d_\varepsilon > 0$, $L_\varepsilon > 0$.
1: **Initialization:** Set $N = \lceil \log_2(d_{\max}/d_\varepsilon) \rceil$ with search upper bound $d_{\max} \triangleq \max\{d_\varepsilon, \|\nabla\ell(\mathbf{x}^0)\|T^2/L_\varepsilon\}$
2: **for** $i = 1, 2, \ldots, N$ **do**
3:      Run UNIXGRAD (see Lemma 5) with initial point $\mathbf{x}^0$, domain diameter $D_i = d_\varepsilon 2^i$, oracle budget $T_i = \lfloor T/(i(1 + \ln N)) \rfloor$, then receive the output $\mathbf{x}^i$
4: **end for**
**Output:** $\mathbf{x}^{\text{out}} = \mathbf{x}^{i_\star}$ with $i_\star = \arg\min_{0 \le i \le N} \ell(\mathbf{x}^i)$.

Moreover, the above convergence rate can be further improved via a *non-uniform* budget allocation. Specifically, with $T_i = \lfloor T/(i(1 + \ln N)) \rfloor$ for $i \in [N]$, we obtain:

$$\mathcal{O}\left(\frac{\bar{L}_\ell \bar{d}_0^2}{T_{i_\star}^2}\right) = \mathcal{O}\left(\frac{\bar{L}_\ell \bar{d}_0^2}{T^2}\left(\log_+ \frac{\bar{d}_0}{d_\varepsilon}\right)^2\right), \qquad (9)$$

which replaces the $d_{\max}$ in Eq. (8) by $\bar{d}_0$. And the dependency on $\log(d_0/d_\varepsilon)$ matches Kreisler et al. (2024, Theorem 1), but without the additional logarithmic factors therein.

**Ensemble.** In the deterministic case, we simply select the candidate with the minimum function value. While in the stochastic setting with access to a noisy function value oracle, we need to allocate more oracle budget to each candidate to obtain an accurate estimate of the function value. We then select $\mathbf{x}^{\text{out}}$ with the smallest estimated function value, ensuring that, for any candidate $\mathbf{x}^i$ where $i \in [N] \cup \{0\}$:

$$\ell(\mathbf{x}^{\text{out}}) - \ell(\mathbf{x}^\star) \le \mathcal{O}\left(\ell(\mathbf{x}^i) - \ell(\mathbf{x}^\star)\right) + \text{ERR}, \qquad (10)$$

where the ensemble error ERR denotes the maximum estimation error among all candidates.

Besides, we emphasize that providing a sharper ensemble error is important, as it depends on the zeroth-order noise $\sigma_\ell$ given in Assumption 2 and thus cannot be simply absorbed into gradient variance terms. To this end, we provide a more refined depiction of the zeroth-order stochastic variance, thereby obtaining a sharper, problem-dependent ensemble error in favorable regimes such as interpolation. More details will be provided in Section 5.1.

To conclude, we present Theorem 1 for deterministic convex and smooth optimization, with a proof sketch. Notably, it is *fully parameter-free* without any knowledge of $L_\ell$ and $d_0$.

**Theorem 1.** *For the deterministic convex optimization, and assume the objective $\ell(\mathbf{x})$ is $L_\ell$-smooth. With any user-specified $L_\varepsilon, d_\varepsilon > 0$, Algorithm 1 enjoys:*

$$\ell(\mathbf{x}^{\text{out}}) - \ell(\mathbf{x}^\star) \le \mathcal{O}\left(\frac{\bar{L}_\ell \bar{d}_0^2}{T^2}\left(\log_+ \frac{\bar{d}_0}{d_\varepsilon}\right)^2\right),$$

*where $\bar{L}_\ell \triangleq \max\{L_\ell, L_\varepsilon\}$, $\bar{d}_0 \triangleq \max\{d_0, d_\varepsilon\}$, $d_0 \triangleq \|\mathbf{x}^0 - \mathbf{x}^\star\|$, and we omit the double logarithmic factor $\log\log(1/L_\varepsilon)$ in notation $\mathcal{O}(\cdot)$.*

**Remark 1.** Theorem 1 is *fully* parameter-free to both $L_\ell$ and $d_0$. We are aware of two related works for the same setting. The first one is Kreisler et al. (2024, Theorem 1), which, besides requiring prior knowledge of $r_\varepsilon \le d_0$, has an additional logarithmic factor of $\log_+^4\left(1 + T\min\{L_\ell d_0^2, Gd_0\}/F_\ell\right)$. The other one is Li & Lan (2025, Corollary 3), an optimal and fully parameter-free result, but its extension to the stochastic setting remains unclear.

*Proof Sketch of Theorem 1.* Since $\mathbf{x}^{\text{out}}$ is the best candidate,

$$\ell(\mathbf{x}^{\text{out}}) - \ell(\mathbf{x}^\star) \le \ell(\mathbf{x}^i) - \ell(\mathbf{x}^\star),$$

for all $i \in [N] \cup \{0\}$. We define the search upper bound $d_{\max} \triangleq \max\{d_\varepsilon, \|\nabla\ell(\mathbf{x}^0)\|T^2/L_\varepsilon\}$, and perform the following case-by-case study for $\bar{d}_0 \triangleq \max\{d_0, d_\varepsilon\}$.

**Case of $\bar{d}_0 > d_{\max}$.** Then $\frac{\bar{L}_\ell \bar{d}_0^2}{T^2} > \|\nabla\ell(\mathbf{x}^0)\|d_0$ implies:

$$\ell(\mathbf{x}^{\text{out}}) - \ell(\mathbf{x}^\star) \le \ell(\mathbf{x}^0) - \ell(\mathbf{x}^\star)$$
$$\le \|\nabla\ell(\mathbf{x}^0)\|d_0 \le \mathcal{O}\left(\frac{\bar{L}_\ell \bar{d}_0^2}{T^2}\right).$$

**Case of $\bar{d}_0 \in [d_\varepsilon, d_{\max}]$.** Let $i_\star = \lceil \log_2(\bar{d}_0/d_\varepsilon) \rceil \in [N]$, its diameter $D_{i_\star} = d_\varepsilon 2^{i_\star}$ that $D_{i_\star}/2 \le \bar{d}_0 \le D_{i_\star}$, and allocated oracle budget is $T_{i_\star} = \lfloor T/(i_\star(1 + \ln N)) \rfloor$. Applying Lemma 5 with $L_\nu = L_\ell, \nu = 1, \delta \to 0_+, \Delta_\ell = 0$:

$$\ell(\mathbf{x}^{\text{out}}) - \ell(\mathbf{x}^\star) \le \ell(\mathbf{x}^{i_\star}) - \ell(\mathbf{x}^\star) \le \mathcal{O}\left(\frac{\bar{L}_\ell D_{i_\star}^2}{T_{i_\star}^2}\right)$$
$$\le \mathcal{O}\left(\frac{\bar{L}_\ell \bar{d}_0^2}{T^2}\left(\log_+ \frac{\bar{d}_0}{d_\varepsilon}\right)^2\right),$$

where $\mathcal{O}(\cdot)$ omits the $\log\log(1/L_\varepsilon)$ factor. Combining the above two cases completes the proof. $\qquad\square$

## 4. Parameter-Free Non-Convex Optimization

In this section, we focus on non-convex optimization, where the objective $\ell(\mathbf{x})$ is $L_\ell$-smooth. In this setup, the best-known rate is achieved by Stochastic Gradient Descent (SGD) (Ghadimi & Lan, 2013), for which the fixed step size must be carefully tuned with respect to the smoothness parameter $L_\ell$, the initial sub-optimality gap $F_\ell$, and the gradient variance bound $\Delta_\ell$, as defined in Section 2.1.

Building on the general grid search methodology and our self-bounding analysis proposed in Section 3, along with an efficient ensemble method from Attia & Koren (2024), we propose a novel *fully parameter-free* method, named GRASP-NC for Non-Convex optimization, as shown in Algorithm 2. Our result matches the optimal rate of *tuned*

---

**Algorithm 2** GRASP-NC

---

**Input:** Oracle budget $T$, initial point $\mathbf{x}^0$, $L_\varepsilon > 0$, $F_\varepsilon > 0$, and confidence level $\delta \in (0, 1/3)$.

1: Independently sample $\mathbf{g}(\cdot)$ at $\mathbf{x}^0$ for $\frac{T}{4}$ times, calculate average $\widehat{\mathbf{g}}^0 = \frac{4}{T} \sum_{t=1}^{T/4} \mathbf{g}_t(\mathbf{x}^0)$

2: $L_{\max} \triangleq \frac{\|\widehat{\mathbf{g}}^0\|^2 T}{F_\varepsilon}$, $F_{\max} \triangleq \frac{\|\widehat{\mathbf{g}}^0\|^2 T}{L_\varepsilon}$, $\Delta_{\max}^2 \triangleq \frac{\|\widehat{\mathbf{g}}^0\|^2 T}{\log \frac{1}{\delta}}$

3: Set the search range for step size: $\eta_{\max} \triangleq \frac{1}{2L_\varepsilon}$, and

$$\eta_{\min} \triangleq \min \left\{ \frac{1}{2 \max\{L_\varepsilon, L_{\max}\}}, \sqrt{\frac{2F_\varepsilon}{\max\{L_\varepsilon, L_{\max}\}\Delta_{\max}^2 T}} \right\}$$

4: $N = \lceil \log_2 \frac{\eta_{\max}}{\eta_{\min}} \rceil$, $K = \lceil \log_2 \frac{1}{\delta} \rceil$, candidate set $\mathcal{S} = \emptyset$

5: **for** $i = 1, 2, \ldots, N$ **do**

6:    Run SGD (see Lemma 2) with initial point $\mathbf{x}_1 = \mathbf{x}^0$, constant step size $\eta_i = \eta_{\min} 2^i$, oracle budget $\lfloor \frac{T}{2N} \rfloor$, then uniformly sample $K$ points from the algorithm's trajectory and add to the candidate set $\mathcal{S}$

7: **end for**

8: Order points in $\mathcal{S}$ as $\mathbf{x}^1, \ldots, \mathbf{x}^{KN}$

9: **for** $i = 1, 2, \ldots, KN$ **do**

10:    Independently sample $\mathbf{g}(\cdot)$ at $\mathbf{x}^i$ for $\frac{T}{4KN}$ times, and calculate average $\widehat{\mathbf{g}}^i = \frac{4KN}{T} \sum_{t=1}^{T/(4KN)} \mathbf{g}(\mathbf{x}^i)$

11: **end for**

**Output:** $\mathbf{x}^{\text{out}} = \mathbf{x}^{i_\star}$ with $i_\star = \arg\min_{0 \le i \le KN} \|\widehat{\mathbf{g}}^i\|$.

---

SGD (Ghadimi & Lan, 2013), up to logarithmic factors, but *without* requiring any information about $L_\ell$, $F_\ell$ and $\Delta_\ell$. Specifically, in Line 1, we sample $\mathbf{g}(\mathbf{x}^0)$ for multiple times to obtain an accurate gradient estimation $\widehat{\mathbf{g}}^0$. In Lines 2-3, we use $\|\widehat{\mathbf{g}}^0\|$ to construct effective boundaries for the problem parameters of $L_\ell$, $F_\ell$ and $\Delta_\ell$. In Line 6, we run SGD with a constant step size $\eta_i = \eta_{\min} 2^i$ in the $i$-th grid. Then in Line 10, we sample each candidate multiple times to obtain accurate estimations. Finally, we select the point with minimum gradient norm as the output.

Below, we provide the guarantee, with proof in Appendix B.

**Theorem 2.** *Under Assumption 1, and assume the objective $\ell(\mathbf{x})$ is $L_\ell$-smooth. Algorithm 2 ensures that, for any $\delta \in (0, 1/3)$, with probability at least $1 - 3\delta$:*

$$\|\nabla \ell(\mathbf{x}^{\text{out}})\|^2 \le \mathcal{O}\left( \sqrt{\frac{\bar{L}_\ell \bar{F}_\ell \Delta_\ell^2}{T} \log_+ \frac{T\|\widehat{\mathbf{g}}^0\|}{L_\varepsilon F_\varepsilon}} \right.$$
$$\left. + \frac{\bar{L}_\ell \bar{F}_\ell + \Delta_\ell^2 (\log \frac{1}{\delta})^2}{T} \log_+ \frac{T\|\widehat{\mathbf{g}}^0\|}{L_\varepsilon F_\varepsilon} \right),$$

*where $\bar{L}_\ell \triangleq \max\{L_\ell, L_\varepsilon\}$, $\bar{F}_\ell \triangleq \max\{\ell(\mathbf{x}^0) - \ell(\mathbf{x}^\star), F_\varepsilon\}$.*

We provide a detailed comparison of our result with those of tuned SGD and Attia & Koren (2024) in Table 1a.

**Remark 2.** In Algorithm 2, the algorithmic inputs $L_\varepsilon$, $F_\varepsilon$ do not need to satisfy any conditions related to $L_\ell$ and $F_\ell$,

validating that our method is *fully* parameter-free. However, compared with tuned SGD, in Theorem 2, the dependency on $L_\ell$ becomes $\max\{L_\ell, L_\varepsilon\}$. This is acceptable because the user can choose a small $L_\varepsilon$, while only affecting the convergence rate by a $\log(1/L_\varepsilon)$ factor.

**Remark 3.** In Line 1 of Algorithm 2, the initial sampling process at $\mathbf{x}^0$ consumes $\frac{T}{4}$ oracle calls. We emphasize that this is mainly for theoretical analysis to ensure a tight convergence rate. In practice, sampling less may not significantly affect performance, as shown in Section 6. Intuitively, in the proof, only if $\mathbf{x}^0$ becomes the best candidate, the rate introduces an estimation error for $\|\widehat{\mathbf{g}}^0\|$, necessitating a sufficiently large sample size. However, in practice, the probability of "$\mathbf{x}^0$ is the best candidate" is small, so reducing the initial sampling may not cause a substantial effect.

## 5. Parameter-Free Convex Optimization

In this section, we investigate convex optimization, where the objective $\ell(\mathbf{x})$ is convex with an unknown smoothness level, as introduced in Section 2.1. Moreover, we aim to achieve "universality" (Nesterov, 2015), i.e., adaptation to an unknown level of (Hölder) smoothness while maintaining the optimal convergence.

**Base Algorithm.** In the stochastic setting, the UNIXGRAD algorithm (Kavis et al., 2019), works in bounded domain with diameter $D$, can ensure a universally optimal convergence rate between the separate cases of smoothness and non-smoothness. In Lemma 5, we prove that UNIXGRAD can achieve the stronger universality to Hölder smoothness:

$$\mathcal{O}\left( \frac{L_\nu D^{1+\nu}}{T^{\frac{1+3\nu}{2}}} + \frac{\Delta_D D \log(1/\delta)}{\sqrt{T}} \right),$$

where $\Delta_D$ is the maximum noise bound defined in Eq. (2). It matches the optimal rate in the deterministic setting (Nesterov, 2015), by recovering the optimal rates in both $L_\ell$-smooth case and $G$-Lipschitz case, respectively.

**Algorithms and Convergence Rates.** In this part, we propose GRASP-C, named GRASP for Convex optimization, in Algorithm 3. Specifically, Line 3 offers two options for the searching upper bound of the initial distance $d_0$:

- OPTION-I (11) for *acceleration-only* in the smooth case;

- OPTION-II (12) for *universality* to Hölder smoothness, but requires a lower bound of the objective value.

In Theorem 3 below, we prove that our *fully parameter-free* algorithm with OPTION-I can achieve near-optimal convergence rate up to logarithmic factors, with an additive ensemble error term. The proof is in Appendix C.1.

**Theorem 3.** *Under Assumptions 1 and 2, and assume the objective $\ell(\mathbf{x})$ is $L_\ell$-smooth. Algorithm 3 using* OPTION-I

**Algorithm 3** GRASP-C

---

**Input:** Oracle budget $T$, initial point $\mathbf{x}^0$, $d_\varepsilon > 0$, $L_\varepsilon > 0$, and $\ell_\varepsilon^\star$ if choose OPTION-II in Eq. (12).

1: Independently sample $\mathbf{g}(\cdot)$ at $\mathbf{x}^0$ for $\frac{T}{8}$ times, calculate average $\widehat{\mathbf{g}}^0 = \frac{8}{T} \sum_{t=1}^{T/8} \mathbf{g}_t(\mathbf{x}^0)$

2: Independently sample $\widetilde{\ell}(\cdot)$ at $\mathbf{x}^0$ for $\frac{T}{8}$ times, calculate average $\widehat{\ell}^0 = \frac{8}{T} \sum_{t=1}^{T/8} \widetilde{\ell}_t(\mathbf{x}^0)$

3: Set $N = \lceil \log_2 \frac{d_{\max}}{d_\varepsilon} \rceil$ with grid search upper bound $d_{\max}$ using one of the following two options:

$$\text{OPTION-I}: \; d_{\max} \triangleq \max\{d_\varepsilon, \frac{\|\widehat{\mathbf{g}}^0\| T^2}{L_\varepsilon}\} \qquad (11)$$

$$\text{OPTION-II}: \; d_{\max} \triangleq \max\{1, d_\varepsilon, \frac{(\widehat{\ell}^0 - \ell_\varepsilon^\star) T^2}{L_\varepsilon}\} \quad (12)$$

4: **for** $i = 1, 2, \ldots, N$ **do**

5:     Run UNIXGRAD (see Lemma 5) with oracle budget $\lfloor \frac{T}{2i(1+\ln N)} \rfloor$, domain diameter $d_\varepsilon 2^i$, initial point $\mathbf{x}^0$, and receive the output $\mathbf{x}^i$ from the algorithm

6:     Independently sample $\widetilde{\ell}(\mathbf{x}^i)$ for $\frac{T}{4N}$ times, calculate average $\widehat{\ell^i} = \frac{4N}{T} \sum_{t=1}^{T/(4N)} \widetilde{\ell}_t(\mathbf{x}^i)$

7: **end for**

**Output:** $\mathbf{x}^{\text{out}} = \mathbf{x}^{i_\star}$ with $i_\star = \arg\min_{0 \leq i \leq N} \widehat{\ell^i}$.

---

in Eq. (11) ensures that, for any $\delta \in (0, 1/2)$, with probability at least $1 - 2\delta$, $\ell(\mathbf{x}^{\text{out}}) - \ell(\mathbf{x}^\star)$ is bounded by:

$$\mathcal{O}\Bigg( \frac{\bar{L}_\ell \bar{d}_0^2}{T^2} \left( \log_+ \frac{\bar{d}_0}{d_\varepsilon} \right)^2 + \frac{\Delta_{2\bar{d}_0} \bar{d}_0}{\sqrt{T}} \left( \log_+ \frac{\bar{d}_0}{d_\varepsilon} \right)^{\frac{1}{2}} \left( \log \frac{1}{\delta} \right)$$

$$+ \text{ERR}_N \left( \frac{T}{N} \right) \Bigg),$$

where $\bar{L}_\ell \triangleq \max\{L_\ell, L_\varepsilon\}$ and $\bar{d}_0 \triangleq \max\{d_0, d_\varepsilon\}$ with any $L_\varepsilon, d_\varepsilon > 0$. $\Delta_{2\bar{d}_0}$ is the maximum noise defined in Eq. (2). $N = \mathcal{O}(\log_+(T\|\widehat{\mathbf{g}}^0\|/(L_\varepsilon d_\varepsilon)))$, and $\text{ERR}_N(\cdot)$ is an ensemble error formally defined in Section 5.1.

Compared to guarantee of UNIXGRAD, our Theorem 3 includes an additional ensemble error ERR term. We will provide its analysis and discussion in the following.

Meanwhile, by setting OPTION-II in Eq. (12), our Algorithm 3 can adapt to Hölder smoothness while maintaining the *fully* parameter-freeness. However, it requires a lower bound of the objective value $\ell_\varepsilon^\star \leq \ell(\mathbf{x}^\star)$ to construct the benchmark convergence rate. We present the convergence rate in the following theorem, with proof in Appendix C.2.

**Theorem 4.** *Under Assumptions 1 and 2, and assume the objective $\ell(\mathbf{x})$ is $(L_\nu, \nu)$-Hölder smooth and $\ell(\mathbf{x}^\star) \geq \ell_\varepsilon^\star$. Algorithm 3 using OPTION-II in Eq. (12) ensures that, for any $\delta \in (0, 1/2)$, with probability at least $1 - 2\delta$, $\ell(\mathbf{x}^{\text{out}}) -$*

*Table 2.* Comparison of ensemble guarantees, i.e. Attia & Koren (2024, Lemma 9) and our Lemma 1, given stochastic function value oracle (Assumption 2). $\text{ERR}_N(M)$ is the ensemble error. '—' denotes no additional condition. $\widetilde{\mathcal{O}}(\cdot)$ omits $\log(N/\delta)$ factors.

| Reference | Condition | $\text{ERR}_N(M)$ |
|---|---|---|
| Attia & Koren (2024) | — | $\widetilde{\mathcal{O}}\left( \frac{\sigma_\ell}{\sqrt{M}} \right)$ |
| **Ours [Lemma 1]** | Eq. (13) | $\widetilde{\mathcal{O}}\left( \frac{\sqrt{V_0}}{\sqrt{M}} + \frac{\sigma_\ell + V_1}{M} \right)$ |

$\ell(\mathbf{x}^\star)$ *is bounded by:*

$$\mathcal{O}\Bigg( \frac{\bar{L}_\nu \bar{d}_0^{1+\nu}}{T^{\frac{1+3\nu}{2}}} \left( \log_+ \frac{\bar{d}_0}{d_\varepsilon} \right)^{\frac{1+3\nu}{2}} + \frac{\Delta_{2\bar{d}_0} \bar{d}_0}{\sqrt{T}} \left( \log_+ \frac{\bar{d}_0}{d_\varepsilon} \right)^{\frac{1}{2}} \left( \log \frac{1}{\delta} \right)$$

$$+ \frac{\sigma_\star \sqrt{\log(1/\delta)}}{\sqrt{T}} + \text{ERR}_N \left( \frac{T}{N} \right) \Bigg),$$

*where $\bar{L}_\nu \triangleq \max\{L_\nu, L_\varepsilon\}$ and $\bar{d}_0 \triangleq \max\{d_0, d_\varepsilon\}$ with any $L_\varepsilon, d_\varepsilon > 0$, $\Delta_{2\bar{d}_0}$ is the maximum noise defined in Eq. (2). $\sigma_\star$ is value noise at $\mathbf{x}^\star$, i.e. $\Pr[|\widetilde{\ell}(\mathbf{x}^\star) - \ell(\mathbf{x}^\star)| \leq \sigma_\star] = 1$. $N = \mathcal{O}(\log_+(T(\widehat{\ell}^0 - \ell_\varepsilon^\star)/(L_\varepsilon d_\varepsilon)))$, and $\text{ERR}_N(\cdot)$ is an ensemble error formally defined in Section 5.1.*

Notably, the assumption of $\ell_\varepsilon^\star \leq \ell(\mathbf{x}^\star)$ is mild and naturally satisfiable in many cases. For example, for non-negative losses, such as the widely used squared losses, cross-entropy losses, hinge losses, etc., we can simply set $\ell_\varepsilon^\star = 0$. We also provide Theorem 5 in appendix to eliminate the need for $\ell_\varepsilon^\star$ while still maintaining universality and full freeness, at the cost of a worse ensemble error of $\mathcal{O}(\text{ERR}_{\sqrt{T}}(\sqrt{T}))$.

Additionally, Theorem 4 introduces an extra term of the zeroth-order noise at $\mathbf{x}^\star$, i.e., $\sigma_\star/\sqrt{T}$. This can be considered small, especially in cases with over-parameterization where $\sigma_\star \approx 0$ (Ma et al., 2018; Liu et al., 2023a).

## 5.1. Sharper Ensemble under Interpolation

We denote by $\text{ERR}_N(M)$ the ensemble error, satisfying that:

$$\ell(\mathbf{x}^{\text{out}}) - \ell(\mathbf{x}^\star) \leq \mathcal{O}(\min_{i \in [N]} \ell(\mathbf{x}_i) - \ell(\mathbf{x}^\star) + \text{ERR}_N(M)),$$

where $N$ candidates are sampled $M$ times, respectively. Comparing function values for ensemble introduces the zeroth-order variance $\sigma_\ell$ to the ensemble error. Specifically, as formally stated in Lemma 7, comparison of function values will lead to a zeroth-order variance $\sigma_\ell/\sqrt{M}$ to the ensemble error, where $M$ is the number of samples.

In the following, we state our new model-ensemble lemma, which leverages a gap-dependent variance bound to achieve a sharper ensemble guarantee. The proof is in Appendix C.3.

**Lemma 1.** *Given a zeroth-order oracle $\widetilde{\ell}(\cdot)$, as formalized in Assumption 2, we further assume that the objective $\ell(\cdot)$*

*admits a minimizer* $\mathbf{x}^\star \in \arg\min_{\mathbf{x}\in\mathcal{X}} \ell(\mathbf{x})$, *and* $\forall \mathbf{x} \in \mathcal{X}$,

$$\text{VAR}[\widetilde{\ell}(\mathbf{x}) \mid \mathbf{x}] \le V_0 + V_1(\ell(\mathbf{x}) - \ell(\mathbf{x}^\star)), \quad (13)$$

*with some constants* $V_0, V_1 \ge 0$. *Given* $\mathbf{x}_1, \ldots, \mathbf{x}_N$, *denote* $\overline{\mathbf{x}} = \arg\min_{i\in[N]} \sum_{j=1}^{M} \widetilde{\ell}_j(\mathbf{x}_i)$. *Then with probability at least* $1 - \delta$, *it holds that*

$$\ell(\overline{\mathbf{x}}) - \ell(\mathbf{x}^\star) \le 3\Big( \min_{i\in[N]} \ell(\mathbf{x}_i) - \ell(\mathbf{x}^\star) \Big)$$
$$+ \frac{\sqrt{32 V_0 \log(2N/\delta)}}{\sqrt{M}} + \frac{(8\sigma_\ell + 24 V_1)\log(2N/\delta)}{3M}.$$

We provide a comparison between our new ensemble error and that from Attia & Koren (2024) in Table 2.

For the assumption $\text{VAR}[\widetilde{\ell}(\mathbf{x}) \mid \mathbf{x}] \le V_0 + V_1(\ell(\mathbf{x}) - \ell(\mathbf{x}^\star))$ in Eq. (13), canonical examples are over-parameterized neural networks and over-parameterized linear models (Ma et al., 2018; Liu et al., 2023a; Evron et al., 2026).

### 5.2. Discussions of Our Results

In this part, we specify the ensemble error in Theorems 3 and 4, and compare our universal rate, Theorem 4, with the best known Kreisler et al. (2024, Theorem 2).

**Additional Ensemble Error.** In Theorems 3 and 4, the ensemble error $\text{ERR}_N(T/N)$, in the worst case, is given by:

$$\text{ERR}_N\left(\frac{T}{N}\right) = \sigma_\ell \sqrt{\frac{\log(N/\delta)}{T/N}} \quad (14)$$

from Attia & Koren (2024, Lemma 9), restated in Lemma 7. Here, the number of candidates $N$ is a logarithmic factor. Notably, although the ensemble error incorporates a zeroth-order variance $\sigma_\ell$, it does not depend on $d_0$, showing that it might be smaller than the variance term like $d_0 \Delta_\ell / \sqrt{T}$.

Moreover, using our new problem-dependent ensemble analysis in Lemma 1, it holds that

$$\text{ERR}_N\left(\frac{T}{N}\right) = \sqrt{\frac{V_0 \log(N/\delta)}{T/N} + \frac{(\sigma_\ell + V_1)\log(N/\delta)}{T/N}}. \quad (15)$$

As $V_0 = \sigma_\ell^2$ and $V_1 = 0$ in the *worst case*, Eq. (15) strictly recovers the original result in Eq. (14). Moreover, Eq. (15) provides a gap-dependent perspective, based on the idea that the variance of the zeroth-order oracle may diminish when approaching the optimum. Intuitively, $V_0$ represents the variance at the optimum, and $V_1$ characterizes how the variance scales with the function value gap of $\ell(\cdot) - \ell(\mathbf{x}^\star)$.

For theoretical considerations, we compare the function values of the base algorithms' last iterates. While in practice, additional information, such as the decreasing behavior of the loss curve along the optimization trajectory, may help select the best candidate more effectively.

**Comparison with Kreisler et al. (2024).** The U-DoG algorithm (Kreisler et al., 2024, Theorem 2) is the best-known result for universal stochastic convex optimization. Given the *true* lower bound $r_\varepsilon$ of $d_0$ and the gradient noise bound function $\widehat{\Delta}(\cdot) : \mathbb{R}^d \to \mathbb{R}_+$ (an upper bound of the true noise function $\Delta(\cdot)$ in Assumption 1), U-DoG ensures:

$$\mathcal{O}\left( C^{\text{U-DoG}}\Big( \min\Big\{ \frac{L_\ell d_0^2}{T^2}, \frac{G d_0}{\sqrt{T}} \Big\} + \frac{d_0 \Delta_{2d_0}}{\sqrt{T}} + \frac{d_0 \widehat{\Delta}_{2d_0}}{T} \Big) \right)$$

with probability at least $1 - \delta$, where the coefficient $C^{\text{U-DoG}}$:

$$\left( \log_+ \frac{d_0}{r_\varepsilon} \right)^2 \left( \log \frac{1}{\delta} \right)^2 \left( \log_+ T \frac{\widehat{\Delta}_{2d_0} + \min\{L_\ell d_0^2, G d_0\}}{\ell(\mathbf{x}_0) - \ell(\mathbf{x}^\star)} \right)^4.$$

U-DoG and our method differ in various aspects, including the methodology, algorithmic conditions, and the final theoretical guarantees. In the following, we discuss these aspects in detail, trying to provide a clear comparison of the advantages and disadvantages between the two methods.

**Methodology:** We apply the grid search method, which is flexible and extendable since we use the base algorithm in a *black-box* manner. Thus, we do not need to dive into the analytical details of UNIXGRAD, and can replace it with any other base algorithm with the same convergence rate. In contrast, U-DoG also applies UNIXGRAD, but in a *white-box* manner to fit the "DoG" analysis (Ivgi et al., 2023) to be free from $d_0$.

**Conditions:** U-DoG requires a lower bound for $d_0$ and the gradient noise bound function $\widehat{\Delta}(\cdot)$. In contrast, our method needs a lower bound for the function value, which is more accessible in some cases. It is worth mentioning that our algorithm does not require any knowledge related to $L_\ell$ and $d_0$, whereas the analysis of U-DoG holds only when given the true lower bound of $d_0$, and a true upper bound of $\Delta(\cdot)$.

**Convergence:** As summarized in Table 1b, our result can adapt to the general Hölder smoothness, whereas U-DoG provides guarantees only for the separate cases of smoothness and non-smoothness. Besides, our coefficients of the common terms, such as the main convergence and the variance terms, are smaller, where a formal comparison is provided in Table 1b. However, our bound includes an ensemble error caused by the nature of the grid search framework, which does not appear in the rate of U-DoG.

## 6. Experiments

In this section, we conduct experiments, aiming to investigate two questions: *(i)* How does our parameter-free method perform compared to the tuned methods? *(ii)* The algorithm

*Table 3.* GRASP-NC. We set $\delta = 0.05$, and vary inputs $(L_\varepsilon, F_\varepsilon)$ in $\{0.001, 0.01, 0.1\} \times \{0.001, 0.01, 0.1\}$, and the initial sampling budget $M$ from $T/4$ to $T/4096$. We report the relative difference in gradient norm $\rho$ between our method and the optimally tuned one.

| Inputs | | $\rho$ | | | | | |
|---|---|---|---|---|---|---|---|
| $L_\varepsilon$ | $F_\varepsilon$ | $M = T/4$ | $M = T/16$ | $M = T/64$ | $M = T/256$ | $M = T/1024$ | $M = T/4096$ |
| 0.001 | 0.001 | 0.1836 | 0.1699 | 0.1699 | 0.1699 | 0.1699 | 0.1699 |
| 0.001 | 0.01 | 0.1468 | 0.1468 | 0.1468 | 0.1468 | 0.1468 | 0.1468 |
| 0.001 | 0.1 | 0.1468 | 0.1468 | 0.1468 | 0.1468 | 0.1468 | 0.1468 |
| 0.01 | 0.001 | 0.2336 | 0.2336 | 0.2336 | 0.2336 | 0.2336 | 0.2336 |
| 0.01 | 0.01 | 0.2094 | 0.1772 | 0.1681 | 0.1772 | 0.2094 | 0.2094 |
| 0.01 | 0.1 | 0.2094 | 0.1572 | 0.1557 | 0.1557 | 0.1583 | 0.1772 |
| 0.1 | 0.001 | 0.2264 | 0.1365 | 0.1365 | 0.1365 | 0.1365 | 0.1657 |
| 0.1 | 0.01 | 0.2023 | 0.1141 | 0.1139 | 0.1139 | 0.1141 | 0.1141 |
| 0.1 | 0.1 | 0.1141 | 0.1139 | 0.1139 | 0.1139 | 0.1139 | 0.1141 |

design requires sufficient initial sampling to ensure theoretical tight convergence; will reducing the initial sampling appropriately have a significant impact on the results? To this end, our experiments conduct two key messages:

(1) Our parameter-free methods obtain competitive convergence compared with optimally tuned methods.

(2) Reducing the initial sample budget has little impact, and may improve the results thanks to the saved budget.

**Setup.** We leverage the ResNet-18 model (He et al., 2016) with an added linear layer, CIFAR-10 dataset (Krizhevsky et al., 2009) (with the batch size of 64), and cross-entropy loss. For the convex case, we use pre-trained ResNet-18 and only update the linear layer. We split the oracle budget $T = 10^4$ into two parts: the initial sampling budget $M$ for determining the search range, and the training budget $T - M$. We then choose the candidate with the best performance (i.e., the minimum averaged gradient norm or loss within a sliding window of the last 100 batches, following a common practical "checkpoint" approach) as the output of our method, denoted as $\mathbf{x}^{\text{out}}$. We vary algorithmic inputs (e.g., $L_\varepsilon, F_\varepsilon$ in Algorithm 2 for non-convex case, and $d_\varepsilon, L_\varepsilon$ in Algorithm 3 for convex case), and the initial sampling budget $M$ from $T/4$ to $T/4096$, to evaluate the sensitivity of our method.

**Measure.** To compare our method with the optimally tuned one, we define *relative difference* $\rho$. Our method outputs $\mathbf{x}^{\text{out}}$. We perform a more fine-grained search to find the best-tuned hyper-parameters for base algorithm (i.e., SGD in Algorithm 2 for non-convex case, and UNIXGRAD in Algorithm 3 for convex case), allocate the *entire* budget $T$ to it, and get $\mathbf{x}^{\text{tuned}}$. Then we define $\rho \triangleq \frac{\|\nabla\ell(\mathbf{x}^{\text{out}})\| - \|\nabla\ell(\mathbf{x}^{\text{tuned}})\|}{\|\nabla\ell(\mathbf{x}^{\text{tuned}})\|}$ for non-convex case, and $\rho \triangleq \frac{\ell(\mathbf{x}^{\text{out}}) - \ell(\mathbf{x}^{\text{tuned}})}{\ell(\mathbf{x}^{\text{tuned}})}$ for convex case. $\rho \geq 0$, and smaller $\rho$ indicates better performance.

**Results of Non-convex Setting.** For Algorithm 2, GRASP-NC, our results are shown in Table 3. We can see that the values of $\rho$ lie between 0.11 and 0.23, meaning the actual impact of the log factor in our convergence rate guarantee

only results in a multiplier of 1.11 to 1.23, justifying the claim that our method achieves competitive performance with the optimally tuned one. Moreover, as the initial sampling budget $M$ decreases, the performance does not noticeably degrade, which indicates that in practice, we can use a smaller $M$ to save oracle calls for training. One intuitive justification for reducing the initial sampling budget is that, as shown in the analysis in Appendix B, the rate incurs an estimation error for $\|\widehat{\mathbf{g}}^0\|$ only when $\mathbf{x}^0$ becomes the best candidate, necessitating a sufficiently large sample size. However, this event is rare in practice, so reducing the initial sampling budget will not cause a substantial effect.

**Results of Convex Setting.** For Algorithm 3, GRASP-C, our results are detailed in Appendix E due to space limit. The results again justify the competitive performance of our methods (with $\rho$ between 0.08 and 0.16), and the insensitivity to reducing the initial sampling budget $M$.

# 7. Conclusion

In this paper, we focus on achieving full parameter-freeness in stochastic optimization. We propose a novel self-bounding analysis technique that effectively derives searching ranges for unknown problem parameters. Our framework is effective in both non-convex and convex settings. In the non-convex setting, we design a fully parameter-free method with the optimal convergence rate up to logarithmic factors. In the convex setting, our method is also competitive with the state-of-the-art results regarding parameter-freeness and convergence. We also provide a new guarantee for model ensembles based on a gap-dependent variance characterization, which may be of independent interest.

We highlight two interesting directions for future work. The first is to apply our framework to other settings, as our self-bounding analysis is quite general and could potentially be extended to parameter-free methods tailored for different scenarios. The second is to develop more effective and efficient ensemble methods to reduce ensemble error and minimize oracle calls, thereby enhancing performance.

## Acknowledgements

This work was supported by NSFC (62361146852), the Fundamental and Interdisciplinary Disciplines Breakthrough Plan of the Ministry of Education of China (No. JYB2025XDXM118), the "111 Center" (No. B26023), and the Fundamental Research Funds for the Central Universities (2026300331). This project has received funding from the European Research Council (ERC) under the European Union's Horizon 2020 research and innovation program (grant agreement No. 101078075). Views and opinions expressed are however those of the author(s) only and do not necessarily reflect those of the European Union or the European Research Council. Neither the European Union nor the granting authority can be held responsible for them. This work received additional support from the Israel Science Foundation (ISF, grant number 3174/23), the Council for Higher Education in Israel under a Moonshot Project, a grant from the Tel Aviv University Center for AI and Data Science (TAD), and a fellowship from the Israeli Council for Higher Education. PZ is grateful to Ashok Cutkosky for initial discussions on grid search.

## Impact Statement

This paper presents work whose goal is to advance the field of machine learning. There are many potential societal consequences of our work, none of which we feel must be specifically highlighted here.

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

# A. Related Work on Parameter-free Stochastic Optimization

In this section, we review related work on parameter-free stochastic optimization, covering the non-convex smooth case in Appendix A.1 and the convex case in Appendix A.2. In Appendix A.3, we discuss previous grid-search methods.

## A.1. Stochastic Non-convex and Smooth Optimization

**Optimal Rates by Tuned SGD.**    In non-convex smooth optimization, for finding a stationary point given $T$ stochastic gradient queries with the variance bounded by $\sigma^2$, Ghadimi & Lan (2013) first showed that SGD with a properly tuned step size achieves the optimal convergence rate of $\mathcal{O}(1/T + \sigma/\sqrt{T})$, provided the algorithm knows the variance bound, the smoothness parameter, and the initial function suboptimality gap. Arjevani et al. (2023) provided a matching lower bound.

**SGD with Adaptive Step Sizes.**    Applying the AdaGrad step size (Duchi et al., 2011), which self-tunes based on gradient statistics, to SGD has been observed to be effective in practice (Tieleman & Hinton, 2012; Kingma & Ba, 2015; Loshchilov & Hutter, 2019). Theoretically, using such adaptive step sizes in SGD can eliminate the need to know certain problem parameters (Li & Orabona, 2019; Ward et al., 2019; Kavis et al., 2022; Faw et al., 2022; Attia & Koren, 2023; Liu et al., 2023b), while still enjoying rate interpolation in the small-noise regime. However, these methods remain suboptimal unless the smoothness parameter and the initial function suboptimality gap are known.

## A.2. Stochastic Convex Optimization

**Parameter-freeness in the Literature.**    Parameter-freeness in convex optimization typically refers to matching the convergence rate of optimally tuned SGD (which requires knowledge of the initial distance to an optimal solution, i.e. $d_0 \triangleq \|\mathbf{x}^0 - \mathbf{x}^\star\|$), achieving $\widetilde{\mathcal{O}}(d_0/\sqrt{T})$ up to unavoidable polylogarithmic factors using $T$ bounded stochastic gradient queries (Mcmahan & Streeter, 2012). The notion of parameter-freeness has gradually focused on being free of $d_0$.

One line of research applies the online-to-batch conversion (Cesa-Bianchi et al., 2004) and focuses on the more general online convex optimization problem, obtaining regret bounds of $\widetilde{\mathcal{O}}(d_0\sqrt{T})$ without requiring prior knowledge of $d_0$ (Mcmahan & Streeter, 2012; Orabona, 2013; McMahan & Orabona, 2014; Orabona, 2014; Orabona & Pál, 2016; Cutkosky & Boahen, 2017; Foster et al., 2017; Orabona & Tommasi, 2017; Cutkosky & Orabona, 2018; Jun & Orabona, 2019; Mhammedi & Koolen, 2020; Zhang & Cutkosky, 2022; Zhang et al., 2022; Jacobsen & Cutkosky, 2022; 2023; Cutkosky & Mhammedi, 2024). For a state-of-the-art result, see Cutkosky & Mhammedi (2024), which further eliminates the need to know the Lipschitz constant of stochastic gradients. As discussed by Cutkosky & Mhammedi (2024, Section 5.2), their in-expectation stochastic convergence rate is optimal up to logarithmic factors (Carmon & Hinder, 2024). Another line of research eschews online learning (Carmon & Hinder, 2022; Ivgi et al., 2023) and achieves high-probability guarantees that improve upon the logarithmic factors of the best online-learning-based results of Cutkosky & Mhammedi (2024).

**Adaptation to Smoothness and Universality.**    It is well known that the convergence rate of GD improves to $\mathcal{O}(d_0^2/T)$ when the objective is smooth, and can be accelerated to $\mathcal{O}(d_0^2/T^2)$ by Nesterov's accelerated gradient (NAG) (Nesterov, 2018). In the stochastic setting with the gradient variance bounded by $\sigma^2$, the accelerated rate becomes $\mathcal{O}(d_0^2/T^2 + \sigma d_0/\sqrt{T})$ (Lan, 2012). The significant performance gap between smooth and non-smooth optimization has motivated the study of *universality* (Nesterov, 2015), aiming for algorithms that adapt to both unknown smooth and non-smooth regimes and achieve optimal rates in each. Several works have studied the interpolation of $(L_\nu, \nu)$-Hölder smoothness settings (Devolder et al., 2014; Nesterov, 2015), where the optimal deterministic rate is $\mathcal{O}(L_\nu d_0^{1+\nu}/T^{(1+3\nu)/2})$ (Nesterov, 2015).

In the stochastic setting, results split between constrained and unconstrained optimization. In the constrained setting, where algorithms are provided the domain diameter $D$, substantial progress toward universality has been made. For example, Kavis et al. (2019) proposed the first universal method in this setting, UNIXGRAD, which attains $\mathcal{O}(D^2/T^2 + \sigma D/\sqrt{T})$ in the smooth case and $\mathcal{O}((G + \sigma)D/\sqrt{T})$ in the non-smooth $G$-Lipschitz case. Further adaptations to Hölder smoothness were developed (Rodomanov et al., 2024; Zhao et al., 2025). UNIXGRAD shows that acceleration can be achieved by combining anytime online-to-batch conversion (Cutkosky, 2019) with online adaptivity to gradient variation (Chiang et al., 2012). Moreover, by strengthening the online algorithm with universality, Zhao et al. (2025) showed that UNIXGRAD also adapts to Hölder smoothness (see, e.g., Lemma 5 in Appendix C.5 for a high-probability guarantee).

For the unconstrained setting, obtaining acceleration via parameter-free gradient-variation regret bounds (Zhao et al., 2026) remains an open problem. Earlier universal works were not accelerated (Levy et al., 2018) until the recent work of Kreisler et al. (2024), which combines the DoG step sizes (Ivgi et al., 2023) with the UNIXGRAD framework (Kavis et al., 2019).

### A.3. Grid Search for Parameter-free Stochastic Optimization

The grid search framework is a practical technique for removing requirements on certain problem parameters. Attia & Koren (2024) and Khaled & Jin (2024) achieved parameter-freeness in both non-convex and convex settings by tuning over ranges of the relevant problem parameters. From a practical perspective, however, true parameter ranges are often unavailable.

## B. Omitted Details for Section 4

*Proof of Theorem 2.* By Lemma 2, given constant step size $\eta \leq 1/(2L_\ell)$, SGD ensures that, with probability at least $1 - \delta$:

$$\frac{1}{T}\sum_{t=1}^{T}\|\nabla\ell(\mathbf{x}_t)\|^2 \leq \frac{4F_\ell}{\eta T} + 4\eta L_\ell \Delta_\ell^2 + \frac{12\Delta_\ell^2 \log\frac{1}{\delta}}{T},$$

where $F_\ell \triangleq \ell(\mathbf{x}_1) - \min_{\mathbf{x}\in\mathbb{R}^d}\ell(\mathbf{x})$. Now with any $L_\varepsilon, \Delta_\varepsilon, F_\varepsilon > 0$, define $\bar{L}_\ell \triangleq \max\{L_\ell, L_\varepsilon\}, \bar{\Delta}_\ell \triangleq \max\{\Delta_\ell, \Delta_\varepsilon\}$ and $\bar{F}_\ell \triangleq \max\{F_\ell, F_\varepsilon\}$. Then SGD with step size $\eta \leq 1/(2\bar{L}_\ell) \leq 1/(2L_\ell)$ ensures that, with probability at least $1 - \delta$:

$$\frac{1}{T}\sum_{t=1}^{T}\|\nabla\ell(\mathbf{x}_t)\|^2 \leq \frac{4\bar{F}_\ell}{\eta T} + 4\eta\bar{L}_\ell\bar{\Delta}_\ell^2 + \frac{12\bar{\Delta}_\ell^2 \log\frac{1}{\delta}}{T} = \mathcal{O}\left(\underbrace{\sqrt{\frac{\bar{L}_\ell\bar{F}_\ell\bar{\Delta}_\ell^2}{T}} + \frac{\bar{L}_\ell\bar{F}_\ell}{T} + \frac{\bar{\Delta}_\ell^2\log\frac{1}{\delta}}{T}}_{\text{TARGET}}\right), \tag{16}$$

where the optimal tuning for the step size $\eta$ is $\min\{1/(2\bar{L}_\ell), \sqrt{\bar{F}_\ell/(\bar{L}_\ell\bar{\Delta}_\ell^2 T)}\}$.

Let $\mathbf{x}^0$ be one of the candidates. By sampling $\mathbf{g}(\mathbf{x}^0)$ for $M_0 \triangleq \lfloor\frac{T}{4}\rfloor$ times and define the average $\widehat{\mathbf{g}}^0 \triangleq \frac{1}{M_0}\sum_{t=1}^{M_0}\mathbf{g}_t(\mathbf{x}^0)$, then applying Lemma 8, with probability at least $1 - \delta$:

$$\|\nabla\ell(\mathbf{x}^0)\|^2 \leq 2\|\widehat{\mathbf{g}}^0\|^2 + 2\|\widehat{\mathbf{g}}^0 - \nabla\ell(\mathbf{x}^0)\|^2 \leq 2\|\widehat{\mathbf{g}}^0\|^2 + \mathcal{O}\left(\frac{(\Delta(\mathbf{x}^0))^2\log\frac{1}{\delta}}{T}\right). \tag{17}$$

**Self-bounding Analysis.** Using Eq. (17) as the comparator, next we consider the condition when target convergence TARGET in Eq. (16) becomes vacuous. For three parameters, $L_\ell, \Delta_\ell$ and $F_\ell$:

- Define $L_{\max} \triangleq \frac{\|\widehat{\mathbf{g}}^0\|^2 T}{F_\varepsilon}$, then $L_\ell > L_{\max}$ implies that $\|\widehat{\mathbf{g}}^0\|^2 < \frac{L_\ell F_\varepsilon}{T} \leq$ TARGET, which makes Eq. (16) vacuous.

- Define $F_{\max} \triangleq \frac{\|\widehat{\mathbf{g}}^0\|^2 T}{L_\varepsilon}$, then $F_\ell > F_{\max}$ implies that $\|\widehat{\mathbf{g}}^0\|^2 < \frac{L_\varepsilon F_\ell}{T} \leq$ TARGET, which makes Eq. (16) vacuous.

- Define $\Delta_{\max}^2 \triangleq \frac{\|\widehat{\mathbf{g}}^0\|^2 T}{\log\frac{1}{\delta}}$, then $\Delta_\ell^2 > \Delta_{\max}^2$ implies that $\|\widehat{\mathbf{g}}^0\|^2 < \frac{\Delta_\ell^2\log\frac{1}{\delta}}{T} \leq$ TARGET, which makes Eq. (16) vacuous.

In conclude, in case of $L_\ell > L_{\max}$ or $F_\ell > F_{\max}$ or $\Delta_\ell^2 > \Delta_{\max}^2$, we have, with probability at least $1 - \delta$:

$$\|\nabla\ell(\mathbf{x}^0)\|^2 \overset{(17)}{\leq} 2\|\widehat{\mathbf{g}}^0\|^2 + \mathcal{O}\left(\frac{(\Delta(\mathbf{x}^0))^2\log\frac{1}{\delta}}{T}\right) \leq \mathcal{O}\left(\sqrt{\frac{\bar{L}_\ell\bar{F}_\ell\bar{\Delta}_\ell^2}{T}} + \frac{\bar{L}_\ell\bar{F}_\ell}{T} + \frac{\bar{\Delta}_\ell^2\log\frac{1}{\delta}}{T} + \frac{(\Delta(\mathbf{x}^0))^2\log\frac{1}{\delta}}{T}\right). \tag{18}$$

In the following, we consider the case of $L_\ell \leq L_{\max}$ and $F_\ell \leq F_{\max}$ and $\Delta_\ell^2 \leq \Delta_{\max}^2$. That is, we have the conditions:

$$\bar{L}_\ell \in [L_\varepsilon, \max\{L_\varepsilon, L_{\max}\}], \quad \bar{F}_\ell \in [F_\varepsilon, \max\{F_\varepsilon, F_{\max}\}], \quad \bar{\Delta}_\ell^2 \in \left[\Delta_\varepsilon^2, \max\{\Delta_\varepsilon^2, \Delta_{\max}^2\}\right],$$

which gives us the grid search range for the optimal step size tuning in target convergence Eq. (16). Specifically, since now for each SGD algorithm we apply a budget $T' \equiv \lfloor\frac{T}{2N}\rfloor$, the optimal tuning $\eta_\star$ is:

$$\eta_\star = \min\left\{\frac{1}{2\bar{L}_\ell}, \sqrt{\frac{2\bar{F}_\ell}{\bar{L}_\ell\bar{\Delta}_\ell^2 T'}}\right\} = \min\left\{\frac{1}{2\bar{L}_\ell}, \sqrt{\frac{2\bar{F}_\ell N}{\bar{L}_\ell\bar{\Delta}_\ell^2 T}}\right\},$$

which has the following lower bound $\eta_{\min}$ and upper bound $\eta_{\max}$:

$$\eta_{\min} = \min\left\{\frac{1}{2\max\{L_\varepsilon, L_{\max}\}}, \sqrt{\frac{2F_\varepsilon}{\max\{L_\varepsilon, L_{\max}\}\cdot\max\{\Delta_\varepsilon^2, \Delta_{\max}^2\}\cdot T}}\right\} \leq \eta_\star \leq \eta_{\max} = \frac{1}{2L_\varepsilon}. \tag{19}$$

We set $\eta_i = \eta_{\min} 2^i$, hence $N = \lceil \log_2 \frac{\eta_{\max}}{\eta_{\min}} \rceil$, and there exists $i_\star = \lceil \log_2 \frac{\eta_\star}{\eta_{\min}} \rceil \in [N]$ such that $\eta_{i_\star}/2 \leq \eta_\star \leq \eta_{i_\star}$, then by Eq. (16), with probability at least $1 - \delta$:

$$\frac{1}{T'} \sum_{t=1}^{T'} \|\nabla \ell(\mathbf{x}_t^{i_\star})\|^2 \leq \mathcal{O} \left( \sqrt{\frac{\bar{L}_\ell \bar{F}_\ell \bar{\Delta}_\ell^2}{T'}} + \frac{\bar{L}_\ell \bar{F}_\ell + \Delta_\ell^2 \log \frac{1}{\delta}}{T'} \right) = \mathcal{O} \left( \sqrt{\frac{\bar{L}_\ell \bar{F}_\ell \bar{\Delta}_\ell^2}{T/N}} + \frac{\bar{L}_\ell \bar{F}_\ell + \Delta_\ell^2 \log \frac{1}{\delta}}{T/N} \right). \quad (20)$$

**Ensemble Error.** By Lemma 4, our ensemble method ensures that, with probability at least $1 - 2\delta$:

$$\|\nabla \ell(\overline{\mathbf{x}})\|^2 \leq 8 \min \left\{ \|\nabla \ell(\mathbf{x}^0)\|^2, \min_{i \in [N]} \frac{1}{T_i} \sum_{t=1}^{T_i} \|\nabla \ell(\mathbf{x}_t^i)\|^2 \right\} + \mathcal{O} \left( \frac{N (\log \frac{N}{\delta})(\log \frac{1}{\delta}) \Delta_\ell^2}{T} \right),$$

therefore, combining Eq. (18) and Eq. (20), we conclude that, with probability at least $1 - 3\delta$:

$$\|\nabla \ell(\overline{\mathbf{x}})\|^2 \leq \mathcal{O} \left( \sqrt{\frac{\bar{L}_\ell \bar{F}_\ell \bar{\Delta}_\ell^2}{T/N}} + \frac{\bar{L}_\ell \bar{F}_\ell + \Delta_\ell^2 \log \frac{1}{\delta}}{T/N} + \frac{(\Delta(\mathbf{x}^0))^2 \log \frac{1}{\delta}}{T} + \frac{N (\log \frac{N}{\delta})(\log \frac{1}{\delta}) \Delta_\ell^2}{T} \right)$$

$$= \mathcal{O} \left( \sqrt{\frac{\bar{L}_\ell \bar{F}_\ell \bar{\Delta}_\ell^2}{T} \log_+ \frac{\eta_{\max}}{\eta_{\min}}} + \frac{\bar{L}_\ell \bar{F}_\ell + \Delta_\ell^2 (\log \frac{1}{\delta})^2}{T} \log_+ \frac{\eta_{\max}}{\eta_{\min}} \right),$$

where we treat double-logarithmic factor $\log(\log \frac{\eta_{\max}}{\eta_{\min}})$ as constant. Finally, by Eq. (19), we have:

$$\frac{\eta_{\max}}{\eta_{\min}} = \mathcal{O} \left( \frac{1}{L_\varepsilon} \cdot \max \left\{ \max\{L_\varepsilon, L_{\max}\}, \sqrt{\frac{1}{F_\varepsilon} \max\{L_\varepsilon, L_{\max}\} \cdot \max\{\Delta_\varepsilon^2, \Delta_{\max}^2\} \cdot T} \right\} \right)$$

$$= \mathcal{O} \left( \max \left\{ \max \left\{ 1, \frac{\|\widehat{\mathbf{g}}^0\|^2 T}{L_\varepsilon F_\varepsilon} \right\}, \sqrt{\frac{1}{F_\varepsilon} \max \left\{ 1, \frac{\|\widehat{\mathbf{g}}^0\|^2 T}{L_\varepsilon F_\varepsilon} \right\} \cdot \max \left\{ \Delta_\varepsilon^2, \frac{\|\widehat{\mathbf{g}}^0\|^2 T}{\log \frac{1}{\delta}} \right\} \cdot T} \right\} \right)$$

$$= \mathcal{O} \left( \text{poly} \left( T, \|\widehat{\mathbf{g}}^0\|, \frac{1}{L_\varepsilon}, \frac{1}{F_\varepsilon}, \Delta_\varepsilon, \frac{1}{\log \frac{1}{\delta}} \right) \right),$$

by which we can set $\Delta_\varepsilon = 0_+$ (appears only in analysis), treat double-logarithmic factor as constant, then finish proof with:

$$\log_+ \frac{\eta_{\max}}{\eta_{\min}} = \mathcal{O} \left( \log_+ \frac{T \|\widehat{\mathbf{g}}^0\|}{L_\varepsilon F_\varepsilon} \right).$$

$\square$

**Lemma 2** (SGD for Non-Convex Smooth Optimization (Lemma 1 of Attia & Koren (2024))). *Under Assumption 1, assume that $\ell(\mathbf{x})$ is $L_\ell$-smooth and lower bounded by $\ell^\star$. The SGD algorithm that updates by $\mathbf{x}_{t+1} = \mathbf{x}_t - \eta \mathbf{g}(\mathbf{x}_t)$ for all $t \in [T]$, when $\eta \leq 1/(2L_\ell)$, ensures that, for any $\delta \in (0, 1)$, with probability at least $1 - \delta$:*

$$\frac{1}{T} \sum_{t=1}^{T} \|\nabla \ell(\mathbf{x}_t)\|^2 \leq \frac{4(\ell(\mathbf{x}_1) - \ell^\star)}{\eta T} + 4\eta L_\ell \Delta_\ell^2 + \frac{12 \Delta_\ell^2 \log \frac{1}{\delta}}{T}.$$

**Lemma 3** (Lemma 2 of Attia & Koren (2024)). *Under Assumption 1, given $N$ candidates $\mathbf{x}_1, \cdots, \mathbf{x}_N$, let $\overline{\mathbf{x}} = \arg\min_{i \in [N]} \| \sum_{j=1}^{M} \mathbf{g}_j(\mathbf{x}_i) \|$, where $\mathbf{g}_j(\mathbf{x}_i)$ means the $j$-th independent gradient query at $\mathbf{x}_i$. Then for any $\delta \in (0, 1)$, with probability at least $1 - \delta$:*

$$\|\nabla \ell(\overline{\mathbf{x}})\|^2 \leq 4 \min_{i \in [N]} \|\nabla \ell(\mathbf{x}_i)\|^2 + \frac{24(1 + 3 \log \frac{N}{\delta}) \Delta_\ell^2}{M}.$$

The following lemma is an abstraction of the ensemble method in Attia & Koren (2024, Theorem 1).

**Lemma 4.** *Under [Assumption 1](), assume that there are total $T$ gradient Oracle budget and $N$ instances of algorithms, and the $i$-th algorithm has a trajectory $\mathbf{x}_1^i, \ldots, \mathbf{x}_{T_i}^i$. Given $\delta \in (0, 1/2)$, for each $i \in [N]$, uniformly at random select $K = \lceil \log_2 \frac{1}{\delta} \rceil$ indices from $[T_i]$, denoted by $\mathcal{K}_i$, and let candidates set $\mathcal{S} = \{\mathbf{x}_k^i : i \in [N], k \in \mathcal{K}_i\}$. Applying ensemble method in [Lemma 3]() to obtain a selection $\overline{\mathbf{x}}$ from $\mathcal{S}$ with $\lfloor M = T/(KN) \rfloor$, ensures that, with probability at least $1 - 2\delta$,*

$$\|\nabla \ell(\overline{\mathbf{x}})\|^2 \le 8 \min_{i \in [N]} \frac{1}{T_i} \sum_{t=1}^{T_i} \|\nabla \ell(\mathbf{x}_t^i)\|^2 + \mathcal{O}\left( \frac{N(\log \frac{N}{\delta})(\log \frac{1}{\delta})\Delta_\ell^2}{T} \right).$$

*Proof of [Lemma 4]().* By Markov's inequality, with probability at least $1/2$, a uniformly at random index $k \in [T_i]$ satisfies:

$$\|\nabla \ell(\mathbf{x}_k^i)\|^2 \le 2\mathbb{E}_t\left[\|\nabla \ell(\mathbf{x}_t^i)\|^2\right] = \frac{2}{T_i} \sum_{t=1}^{T_i} \|\nabla \ell(\mathbf{x}_t^i)\|^2. \tag{21}$$

Then as we sample $K \triangleq \lceil \log_2 \frac{1}{\delta} \rceil$ random indices, with probability at least $1 - \delta$, we add at least one point with the guarantee [Eq. (21)]() to the candidate set $\mathcal{S}$. Therefore, there are total $KN$ candidates in $\mathcal{S}$. Applying [Lemma 3]() with $M = \lfloor T/(KN) \rfloor$, then, with probability at least $1 - 2\delta$, the selected candidate $\overline{\mathbf{x}}$ satisfies:

$$\|\nabla \ell(\overline{\mathbf{x}})\|^2 \le 4 \min_{\mathbf{x} \in \mathcal{S}} \|\nabla \ell(\mathbf{x})\|^2 + \frac{24KN(1 + 3\log \frac{KN}{\delta})\Delta_\ell^2}{T} \overset{(21)}{\le} 8 \min_{i \in [N]} \frac{1}{T_i} \sum_{t=1}^{T_i} \|\nabla \ell(\mathbf{x}_t^i)\|^2 + \mathcal{O}\left( \frac{N(\log \frac{N}{\delta})(\log \frac{1}{\delta})\Delta_\ell^2}{T} \right),$$

which completes the proof. $\qquad\square$

## C. Omitted Details for Section [5]()

In this section, we provide the omitted details for [Section 5](), including the proofs of Theorems [3](), [4](), [Lemma 1](), a universal convergence rate with less prior knowledge but a worse $\mathcal{O}(\sigma_\ell/T^{1/4})$ ensemble error in [Appendix C.4](). Finally, in [Appendix C.5](), we provide a self-contained analysis of UNIXGRAD.

### C.1. Proof of [Theorem 3]()

*Proof.* We independently sample $\mathbf{g}(\mathbf{x}^0)$ for $\frac{T}{8}$ times, define $\widehat{\mathbf{g}}^0 = \frac{8}{T} \sum_{t=1}^{T/8} \mathbf{g}_t(\mathbf{x}^0)$ where $\mathbf{g}_t(\mathbf{x}^0)$ means the $t$-th gradient query on $\mathbf{x}^0$. By [Lemma 8](), with probability at least $1 - \delta$:

$$\|\widehat{\mathbf{g}}^0 - \nabla \ell(\mathbf{x}^0)\| \le \mathcal{O}\left( \frac{\Delta(\mathbf{x}^0)\sqrt{\log_+ \frac{1}{\delta}}}{\sqrt{T}} \right).$$

Therefore, with probability at least $1 - \delta$:

$$\ell(\mathbf{x}^0) - \ell(\mathbf{x}^\star) \le \|\nabla \ell(\mathbf{x}^0)\| d_0 \le \left(\|\widehat{\mathbf{g}}^0\| + \|\widehat{\mathbf{g}}^0 - \nabla \ell(\mathbf{x}^0)\|\right) d_0 \le \|\widehat{\mathbf{g}}^0\| d_0 + \mathcal{O}\left( \frac{\Delta(\mathbf{x}^0) d_0 \sqrt{\log_+ \frac{1}{\delta}}}{\sqrt{T}} \right). \tag{22}$$

Let $\text{ERR}_N(M)$ be ensemble error given $N$ candidates with $M$ oracle budget for each, with probability at least $1 - \delta$, as stated in [Section 5.1](). That means with probability at least $1 - \delta$, for any $0 \le i_\star \le N$,

$$\ell(\mathbf{x}^{\text{out}}) - \ell(\mathbf{x}^\star) \le \ell(\mathbf{x}^{i_\star}) - \ell(\mathbf{x}^\star) + \mathcal{O}\left( \text{ERR}_N\left(\frac{T}{N}\right) \right). \tag{23}$$

Next we perform the following case-by-case study for $\bar{d}_0 \triangleq \max\{d_0, d_\varepsilon\}$ where $d_0 \triangleq \|\mathbf{x}^0 - \mathbf{x}^\star\|$. The grid search upper bound is $d_{\max} \triangleq \max\{d_\varepsilon, \|\widehat{\mathbf{g}}^0\| T^2/L_\varepsilon\}$, hence $N = \mathcal{O}\left( \log_+ \frac{T\|\widehat{\mathbf{g}}^0\|}{L_\varepsilon d_\varepsilon} \right)$. And we define $\bar{L}_\ell \triangleq \max\{L_\ell, L_\varepsilon\}$.

**Case of $\bar{d}_0 > d_{\max}$.** Then $\frac{\bar{L}_\ell \bar{d}_0^2}{T^2} \geq \|\widehat{\mathbf{g}}^0\| d_0$. Let $i_\star = 0$, with probability at least $1 - 2\delta$:

$$\ell(\mathbf{x}^{\mathrm{out}}) - \ell(\mathbf{x}^\star) \overset{(23)}{\leq} \ell(\mathbf{x}^0) - \ell(\mathbf{x}^\star) + \mathcal{O}\left(\mathrm{ERR}_N\left(\frac{T}{N}\right)\right)$$

$$\overset{(22)}{\leq} \|\widehat{\mathbf{g}}^0\| d_0 + \mathcal{O}\left(\frac{\Delta(\mathbf{x}^0) d_0 \sqrt{\log_+ \frac{1}{\delta}}}{\sqrt{T}} + \mathrm{ERR}_N\left(\frac{T}{N}\right)\right)$$

$$\leq \mathcal{O}\left(\frac{\bar{L}_\ell \bar{d}_0^2}{T^2} + \frac{\Delta(\mathbf{x}^0) d_0 \sqrt{\log_+ \frac{1}{\delta}}}{\sqrt{T}} + \mathrm{ERR}_N\left(\frac{T}{N}\right)\right).$$

**Case of $\bar{d}_0 \in [d_\varepsilon, d_{\max}]$.** Let $i_\star = \lceil \log_2 \frac{\bar{d}_0}{d_\varepsilon} \rceil$, then $D_{i_\star}/2 \leq \bar{d}_0 \leq D_{i_\star} = d_\varepsilon 2^{i_\star}$, and allocated oracle budget $T_{i_\star} = \lfloor \frac{T}{2^{i_\star}(1+\ln N)} \rfloor$. Applying Lemma 5 ($L_\nu = L_\ell, \nu = 1$), with probability at least $1 - 2\delta$:

$$\ell(\mathbf{x}^{\mathrm{out}}) - \ell(\mathbf{x}^\star) \overset{(23)}{\leq} \ell(\mathbf{x}^{i_\star}) - \ell(\mathbf{x}^\star) + \mathcal{O}\left(\mathrm{ERR}_N\left(\frac{T}{N}\right)\right)$$

$$\leq \mathcal{O}\left(\frac{L_\ell D_{i_\star}^2}{T_{i_\star}^2} + \frac{\theta_{T_{i_\star},\delta}\Delta_{D_{i_\star}} D_{i_\star}}{\sqrt{T_{i_\star}}} + \mathrm{ERR}_N\left(\frac{T}{N}\right)\right)$$

$$= \mathcal{O}\left(\frac{L_\ell \bar{d}_0^2}{T^2}\left(\log_+ \frac{\bar{d}_0}{d_\varepsilon}\right)^2 + \frac{\Delta_{2\bar{d}_0} \bar{d}_0}{\sqrt{T}}\left(\log_+ \frac{\bar{d}_0}{d_\varepsilon}\right)^{\frac{1}{2}}\left(\log \frac{1}{\delta}\right) + \mathrm{ERR}_N\left(\frac{T}{N}\right)\right),$$

where $\theta_{T,\delta} \triangleq \log \frac{\log T}{\delta}$, and $\mathcal{O}(\cdot)$ omits double logarithmic factors.

Combining the above two cases, with probability at least $1 - 2\delta$:

$$\ell(\mathbf{x}^{\mathrm{out}}) - \ell(\mathbf{x}^\star) \leq \mathcal{O}\left(\frac{\bar{L}_\ell \bar{d}_0^2}{T^2}\left(\log_+ \frac{\bar{d}_0}{d_\varepsilon}\right)^2 + \frac{\Delta_{2\bar{d}_0} \bar{d}_0}{\sqrt{T}}\left(\log_+ \frac{\bar{d}_0}{d_\varepsilon}\right)^{\frac{1}{2}}\left(\log \frac{1}{\delta}\right) + \mathrm{ERR}_N\left(\frac{T}{N}\right)\right),$$

where $N = \mathcal{O}\left(\log_+ \frac{T\|\widehat{\mathbf{g}}^0\|}{L_\varepsilon d_\varepsilon}\right)$. $\qquad\square$

### C.2. Proof of Theorem 4

*Proof.* Let $\mathrm{ERR}_N(M)$ be ensemble error given $N$ candidates with $M$ oracle budget for each, with probability at least $1 - \delta$, as stated in Section 5.1. That means with probability at least $1 - \delta$, for any $0 \leq i_\star \leq N$,

$$\ell(\mathbf{x}^{\mathrm{out}}) - \ell(\mathbf{x}^\star) \leq \ell(\mathbf{x}^{i_\star}) - \ell(\mathbf{x}^\star) + \mathcal{O}\left(\mathrm{ERR}_N\left(\frac{T}{N}\right)\right). \tag{24}$$

Next we perform the following case-by-case study for $\bar{d}_0 \triangleq \max\{d_0, d_\varepsilon\}$ where $d_0 \triangleq \|\mathbf{x}^0 - \mathbf{x}^\star\|$. The grid search upper bound is $d_{\max} \triangleq \max\{1, d_\varepsilon, (\widehat{\ell}^0 - \ell_\varepsilon^\star)T^2/L_\varepsilon\}$, hence $N = \mathcal{O}\left(\log_+ \frac{T(\widehat{\ell}^0 - \ell_\varepsilon^\star)}{L_\varepsilon d_\varepsilon}\right)$. And we define $\bar{L}_\nu \triangleq \max\{L_\nu, L_\varepsilon\}$.

**Case of $\bar{d}_0 > d_{\max}$.** Then $\frac{\bar{L}_\nu \bar{d}_0^{1+\nu}}{T^{\frac{1+3\nu}{2}}} \geq \frac{L_\varepsilon \bar{d}_0}{T^2} > \widehat{\ell}^0 - \ell_\varepsilon^\star$. Let $i_\star = 0$, with probability at least $1 - 2\delta$,

$$\ell(\mathbf{x}^{\mathrm{out}}) - \ell(\mathbf{x}^\star) \overset{(24)}{\leq} \ell(\mathbf{x}^0) - \ell(\mathbf{x}^\star) + \mathcal{O}\left(\mathrm{ERR}_N\left(\frac{T}{N}\right)\right)$$

$$\leq \widehat{\ell}^0 - \ell_\varepsilon^\star + \mathcal{O}\left(\frac{\sigma_0\sqrt{\log(1/\delta)}}{\sqrt{T}} + \mathrm{ERR}_N\left(\frac{T}{N}\right)\right)$$

$$\leq \mathcal{O}\left(\frac{\bar{L}_\nu \bar{d}_0^{1+\nu}}{T^{\frac{1+3\nu}{2}}} + \frac{\sigma_0\sqrt{\log(1/\delta)}}{\sqrt{T}} + \mathrm{ERR}_N\left(\frac{T}{N}\right)\right),$$

where the second inequality uses Lemma 7 with $N = 1$, $M = \frac{T}{8}$, and define $\sigma_0$ as the maximum function value noise at $\mathbf{x}^0$, i.e. $\Pr[|\ell(\mathbf{x}^0) - \widetilde{\ell}(\mathbf{x}^0)| \leq \sigma_0] = 1$. Moreover, we can decompose $\sigma_0$ by:

$$|\ell(\mathbf{x}^0) - \widetilde{\ell}(\mathbf{x}^0)| \leq |(\ell(\mathbf{x}^0) - \widetilde{\ell}(\mathbf{x}^0)) - (\ell(\mathbf{x}^\star) - \widetilde{\ell}(\mathbf{x}^\star))| + |\ell(\mathbf{x}^\star) - \widetilde{\ell}(\mathbf{x}^\star)| \leq d_0 \Delta_{d_0} + |\ell(\mathbf{x}^\star) - \widetilde{\ell}(\mathbf{x}^\star)|,$$

where we use the fact that $\|\nabla[\ell(\mathbf{x}) - \widetilde{\ell}(\mathbf{x})]\| = \|\nabla \ell(\mathbf{x}) - \mathbf{g}(\mathbf{x})\| \leq \Delta(\mathbf{x})$, as defined in Section 2.1. Therefore,

$$\frac{\sigma_0 \sqrt{\log(1/\delta)}}{\sqrt{T}} \leq \frac{d_0 \Delta_{d_0} \sqrt{\log(1/\delta)}}{\sqrt{T}} + \frac{\sigma_\star \sqrt{\log(1/\delta)}}{\sqrt{T}}.$$

**Case of $\bar{d}_0 \in [d_\varepsilon, d_{\max}]$.** Let $i_\star = \lceil \log_2 \frac{\bar{d}_0}{d_\varepsilon} \rceil$, then $D_{i_\star}/2 \leq \bar{d}_0 \leq D_{i_\star} = d_\varepsilon 2^{i_\star}$, and allocated oracle budget $T_{i_\star} = \lfloor \frac{T}{2i_\star(1+\ln N)} \rfloor$. Applying Lemma 5, with probability at least $1 - 2\delta$,

$$\ell(\mathbf{x}^{\text{out}}) - \ell(\mathbf{x}^\star) \overset{(24)}{\leq} \ell(\mathbf{x}^{i_\star}) - \ell(\mathbf{x}^\star) + \mathcal{O}\left( \text{ERR}_N\left( \frac{T}{N} \right) \right)$$

$$\leq \mathcal{O}\left( \frac{L_\nu D_{i_\star}^{1+\nu}}{T_{i_\star}^{\frac{1+3\nu}{2}}} + \frac{\theta_{T_{i_\star},\delta} \Delta_{D_{i_\star}} D_{i_\star}}{\sqrt{T_{i_\star}}} + \text{ERR}_N\left( \frac{T}{N} \right) \right)$$

$$= \mathcal{O}\left( \frac{L_\nu \bar{d}_0^{1+\nu}}{T^{\frac{1+3\nu}{2}}} \left( \log_+ \frac{\bar{d}_0}{d_\varepsilon} \right)^{\frac{1+3\nu}{2}} + \frac{\theta_{T,\delta} \Delta_{2\bar{d}_0} \bar{d}_0}{\sqrt{T}} \left( \log_+ \frac{\bar{d}_0}{d_\varepsilon} \right)^{\frac{1}{2}} + \text{ERR}_N\left( \frac{T}{N} \right) \right),$$

where $\theta_{T,\delta} \triangleq \log \frac{\log T}{\delta}$, and $\mathcal{O}(\cdot)$ omits double logarithmic factors.

Combining the above two cases, with probability at least $1 - 2\delta$:

$$\ell(\mathbf{x}^{\text{out}}) - \ell(\mathbf{x}^\star) \leq \mathcal{O}\left( \frac{\bar{L}_\nu \bar{d}_0^{1+\nu}}{T^{\frac{1+3\nu}{2}}} \left( \log_+ \frac{\bar{d}_0}{d_\varepsilon} \right)^{\frac{1+3\nu}{2}} + \frac{\Delta_{2\bar{d}_0} \bar{d}_0}{\sqrt{T}} \left( \log_+ \frac{\bar{d}_0}{d_\varepsilon} \right)^{\frac{1}{2}} \left( \log \frac{1}{\delta} \right) + \frac{\sigma_\star \sqrt{\log(1/\delta)}}{\sqrt{T}} + \text{ERR}_N\left( \frac{T}{N} \right) \right),$$

where $N = \mathcal{O}\left( \log_+ \frac{T(\widehat{\ell}^0 - \ell_\varepsilon^\star)}{L_\varepsilon d_\varepsilon} \right)$. $\square$

### C.3. Proof of Lemma 1

*Proof.* Let $i \in [N]$. Using a two-sided Bernstein's inequality (e.g., using Proposition 2.14 of Wainwright (2019)), for any $t > 0$, denoting by $\ell^i = \ell(\mathbf{x}_i)$ and $\widehat{\ell^i} = \frac{1}{M} \sum_{j=1}^{M} \widetilde{\ell}_j(\mathbf{x}_i)$, it holds that

$$\Pr\left[ |\widehat{\ell^i} - \ell^i| \geq t \right] \leq 2 \exp\left( -\frac{Mt^2}{2\text{VAR}[\widetilde{\ell}(\mathbf{x}_i) \mid \mathbf{x}_i] + \frac{2}{3}\sigma_\ell t} \right).$$

Hence, for any $\delta \in (0,1)$, setting $t = \sqrt{2\text{VAR}[\widetilde{\ell}(\mathbf{x}_i) \mid \mathbf{x}_i] \log(2/\delta)/M} + \frac{2}{3M}\sigma_\ell \log(2/\delta)$, with probability at least $1 - \delta$,

$$|\widehat{\ell^i} - \ell^i| < \frac{\sqrt{2\text{VAR}[\widetilde{\ell}(\mathbf{x}_i) \mid \mathbf{x}_i] \log(2/\delta)}}{\sqrt{M}} + \frac{2\sigma_\ell \log(2/\delta)}{3M}.$$

Thus, performing a union bound over $i \in [N]$ and setting $\delta = \delta'/N$, with probability at least $1 - \delta'$, for all $i \in [N]$,

$$|\widehat{\ell^i} - \ell^i| < \sqrt{\frac{2V_0 \log(2N/\delta')}{M} + \frac{4V_1^2 \log^2(2N/\delta')}{M^2} + \frac{(\ell^i - \ell^\star)^2}{4}} + \frac{2\sigma_\ell \log(2N/\delta')}{3M}.$$

$$\leq \frac{1}{2}(\ell^i - \ell^\star) + \frac{\sqrt{2V_0 \log(2N/\delta')}}{\sqrt{M}} + \frac{(2\sigma_\ell + 6V_1) \log(2N/\delta')}{3M}.$$

---

**Algorithm 4** GRASP-C without Lower Bound of Function Value

---

**Input:** Oracle budget $T$, initial point $\mathbf{x}^0$, $d_\varepsilon > 0$.
1: Set $N = \lceil \sqrt{T} \rceil$
2: Independently sample $\widetilde{\ell}(\mathbf{x}^0)$ for $\frac{T}{2(N+1)}$ times, calculate average $\widehat{\ell^0} = \frac{2(N+1)}{T} \sum_{t=1}^{T/(2(N+1))} \widetilde{\ell}_t(\mathbf{x}^0)$
3: **for** $i = 1, 2, \ldots, N$ **do**
4:     Run UNIXGRAD (see Lemma 5) with initial point $\mathbf{x}_1 = \mathbf{x}^0$, domain diameter $D_i = d_\varepsilon 2^i$, Oracle budget $T_i = \lfloor \frac{T}{i^2 \pi^2/3} \rfloor$, and receive the output $\mathbf{x}^i$ from the algorithm
5:     Independently sample $\widetilde{\ell}(\mathbf{x}^i)$ for $\frac{T}{2(N+1)}$ times, calculate average $\widehat{\ell^i} = \frac{2(N+1)}{T} \sum_{t=1}^{T/(2(N+1))} \widetilde{\ell}_t(\mathbf{x}^i)$
6: **end for**
**Output:** $\mathbf{x}^{\text{out}} = \mathbf{x}^{i_\star}$ with $i_\star = \arg\min_{0 \le i \le N} \widehat{\ell^i}$.

---

The rest of the argument will be conditioned on the above high-probability event, under which it holds that

$$\widehat{\ell^i} - \ell^\star \le \frac{3}{2}(\ell^i - \ell^\star) + \frac{\sqrt{2V_0 \log(2N/\delta')}}{\sqrt{M}} + \frac{(2\sigma_\ell + 6V_1)\log(2N/\delta')}{3M}$$

and

$$\ell^i - \ell^\star \le 2(\widehat{\ell^i} - \ell^\star) + \frac{\sqrt{8V_0 \log(2N/\delta')}}{\sqrt{M}} + \frac{(4\sigma_\ell + 12V_1)\log(2N/\delta')}{3M}.$$

Thus, for $\widehat{i} \in \arg\min_{i \in [N]} \widehat{\ell^i}$ and $i_\star \in \arg\min_{i \in [N]} \ell^i$,

$$\begin{aligned}
\ell^{\widehat{i}} - \ell^\star &\le 2(\widehat{\ell^{\widehat{i}}} - \ell^\star) + \frac{\sqrt{8V_0 \log(2N/\delta')}}{\sqrt{M}} + \frac{(4\sigma_\ell + 12V_1)\log(2N/\delta')}{3M} \\
&\le 2(\widehat{\ell^{i_\star}} - \ell^\star) + \frac{\sqrt{8V_0 \log(2N/\delta')}}{\sqrt{M}} + \frac{(4\sigma_\ell + 12V_1)\log(2N/\delta')}{3M} \\
&\le 3(\ell^{i_\star} - \ell^\star) + \frac{\sqrt{32V_0 \log(2N/\delta')}}{\sqrt{M}} + \frac{(8\sigma_\ell + 24V_1)\log(2N/\delta')}{3M},
\end{aligned}$$

which finishes the proof. $\qquad\square$

## C.4. Universal Convergence with Less Prior Knowledge but Worse Ensemble Error

In this part, we prove that *without* the prior knowledge of the lower bound $\ell(\mathbf{x}^\star) \ge \ell_\varepsilon^\star$, we can still achieve universal convergence to Hölder smoothness, while with a larger ensemble error due to the increased number of base algorithms, which leads to an $\widetilde{\mathcal{O}}(\sigma_\ell/T^{1/4})$ term in the worst case.

**Theorem 5.** *Under Assumptions 1 and 2, and assume the objective $\ell(\mathbf{x})$ is $(L_\nu, \nu)$-Hölder smooth, Algorithm 4 ensures that, with probability at least $1 - 2\delta$:*

$$\ell(\mathbf{x}^{\text{out}}) - \ell(\mathbf{x}^\star) \le \mathcal{O}\left( \frac{L_\nu \bar{d}_0^{1+\nu}}{T^{\frac{1+3\nu}{2}}} \left(\log_+ \frac{\bar{d}_0}{d_\varepsilon}\right)^{1+3\nu} + \frac{\Delta_{2\bar{d}_0} \bar{d}_0}{\sqrt{T}} \left(\log_+ \frac{\bar{d}_0}{d_\varepsilon}\right) \left(\log \frac{1}{\delta}\right) + \text{ERR}_{\sqrt{T}}\left(\sqrt{T}\right) \right),$$

*where $\bar{d}_0 \triangleq \max\{d_0, d_\varepsilon\}$, $\Delta_{2\bar{d}_0}$ is the maximum noise defined in Eq. (2), ERR is the ensemble error specified in Section 5.1.*

*Proof.* Set the total number of base algorithms to be $N = \lceil \sqrt{T} \rceil$, and oracle budget $T_i = \lfloor \frac{T}{i^2 \pi^2/3} \rfloor$ for the $i$-th base algorithm. We can verify that $\sum_{i=1}^N T_i \le \frac{T}{\pi^2/3} \sum_{i=1}^\infty \frac{1}{i^2} = \frac{T}{2}$. Let $\text{ERR}_N(M)$ be ensemble error given $N$ candidates with $M$ oracle budget for each, with probability at least $1 - \delta$, as stated in Section 5.1. That means with probability at least $1 - \delta$, for any $0 \le i_\star \le N$,

$$\ell(\mathbf{x}^{\text{out}}) - \ell(\mathbf{x}^\star) \le \ell(\mathbf{x}^{i_\star}) - \ell(\mathbf{x}^\star) + \mathcal{O}\left( \text{ERR}_N\left(\frac{T}{N}\right) \right). \tag{25}$$

Now we perform the following case-by-case study for $\bar{d}_0 \triangleq \max\{d_0, d_\varepsilon\}$, with $d_0 \triangleq \|\mathbf{x}^0 - \mathbf{x}^\star\|$ and any $d_\varepsilon > 0$.

---

**Algorithm 5** UNIXGRAD (Kavis et al., 2019)

---

**Input:** Oracle budget $2T$, domain diameter $D$, initial point $\widehat{\mathbf{x}}_1$, weight $\alpha_t = t$.
1: **for** $t = 1, 2, \ldots, T$ **do**
2:      Set step size $\eta_t = \frac{2D}{\sqrt{1 + \sum_{s=1}^{t-1} \alpha_s^2 \|\mathbf{g}(\overline{\mathbf{x}}_s) - \mathbf{g}(\widetilde{\mathbf{x}}_s)\|^2}}$
3:      Update $\mathbf{x}_t = \arg\min_{\mathbf{x} \in \mathcal{X}} \langle \alpha_t \mathbf{g}(\widetilde{\mathbf{x}}_t), \mathbf{x} \rangle + \frac{1}{2\eta_t} \|\mathbf{x} - \widehat{\mathbf{x}}_t\|^2$ with $\widetilde{\mathbf{x}}_t = \frac{1}{\sum_{s=1}^t \alpha_s} (\alpha_t \widehat{\mathbf{x}}_t + \sum_{s=1}^{t-1} \alpha_s \mathbf{x}_s)$
4:      Update $\widehat{\mathbf{x}}_{t+1} = \arg\min_{\mathbf{x} \in \mathcal{X}} \langle \alpha_t \mathbf{g}(\overline{\mathbf{x}}_t), \mathbf{x} \rangle + \frac{1}{2\eta_t} \|\mathbf{x} - \widehat{\mathbf{x}}_t\|^2$ with $\overline{\mathbf{x}}_t = \frac{1}{\sum_{s=1}^t \alpha_s} (\alpha_t \mathbf{x}_t + \sum_{s=1}^{t-1} \alpha_s \mathbf{x}_s)$
5: **end for**
**Output:** $\overline{\mathbf{x}}_T$.

---

**Case of $\bar{d}_0 > d_\varepsilon 2^N$.** Then $\log_2 \frac{\bar{d}_0}{d_\varepsilon} > N \geq \sqrt{T}$. Let $i_\star = 0$, with probability at least $1 - \delta$:

$$\ell(\mathbf{x}^{\text{out}}) - \ell(\mathbf{x}^\star) \overset{(25)}{\leq} \ell(\mathbf{x}^0) - \ell(\mathbf{x}^\star) + \mathcal{O}\left(\text{ERR}_N\left(\frac{T}{N}\right)\right) \leq L_\nu d_0^{1+\nu} + \mathcal{O}\left(\text{ERR}_N\left(\frac{T}{N}\right)\right)$$

$$\leq \mathcal{O}\left(\frac{L_\nu d_0^{1+\nu}}{T^{\frac{1+3\nu}{2}}} \left(\log_+ \frac{\bar{d}_0}{d_\varepsilon}\right)^{1+3\nu} + \mathcal{O}\left(\text{ERR}_N\left(\frac{T}{N}\right)\right)\right),$$

where the second inequality is because $\nabla \ell(\mathbf{x}^\star) = \mathbf{0}$ and $\ell(\mathbf{x}) - \ell(\mathbf{x}^\star) \leq \|\nabla \ell(\mathbf{x}) - \nabla \ell(\mathbf{x}^\star)\| \|\mathbf{x} - \mathbf{x}^\star\| \leq L_\nu \|\mathbf{x} - \mathbf{x}^\star\|^{1+\nu}$ for all $\mathbf{x} \in \mathcal{X}$, and the third inequality is because $\log_2 \frac{\bar{d}_0}{d_\varepsilon} \geq \sqrt{T}$ by assumption.

**Case of $\bar{d}_0 \in [d_\varepsilon, d_\varepsilon 2^N]$.** Let $i_\star = \lceil \log_2 \frac{\bar{d}_0}{d_\varepsilon} \rceil$, then $D_{i_\star}/2 \leq \bar{d}_0 \leq D_{i_\star} = d_\varepsilon 2^{i_\star}$, and allocated oracle budget $T_{i_\star} = \lfloor \frac{T}{i_\star^2 \pi^2/3} \rfloor$. Applying Lemma 5, with probability at least $1 - 2\delta$,

$$\ell(\mathbf{x}^{\text{out}}) - \ell(\mathbf{x}^\star) \overset{(25)}{\leq} \ell(\mathbf{x}^{i_\star}) - \ell(\mathbf{x}^\star) + \mathcal{O}\left(\text{ERR}_N\left(\frac{T}{N}\right)\right)$$

$$\leq \mathcal{O}\left(\frac{L_\nu D_{i_\star}^{1+\nu}}{T_{i_\star}^{\frac{1+3\nu}{2}}} + \frac{\theta_{T_{i_\star},\delta} \Delta_{D_{i_\star}} D_{i_\star}}{\sqrt{T_{i_\star}}} + \text{ERR}_N\left(\frac{T}{N}\right)\right)$$

$$= \mathcal{O}\left(\frac{L_\nu \bar{d}_0^{1+\nu}}{T^{\frac{1+3\nu}{2}}} \left(\log_+ \frac{\bar{d}_0}{d_\varepsilon}\right)^{1+3\nu} + \frac{\Delta_{2\bar{d}_0} \bar{d}_0}{\sqrt{T}} \left(\log_+ \frac{\bar{d}_0}{d_\varepsilon}\right) \left(\log \frac{1}{\delta}\right) + \text{ERR}_N\left(\frac{T}{N}\right)\right).$$

Combining the above two cases, with probability at least $1 - 2\delta$:

$$\ell(\mathbf{x}^{\text{out}}) - \ell(\mathbf{x}^\star) \leq \mathcal{O}\left(\frac{L_\nu \bar{d}_0^{1+\nu}}{T^{\frac{1+3\nu}{2}}} \left(\log_+ \frac{\bar{d}_0}{d_\varepsilon}\right)^{1+3\nu} + \frac{\Delta_{2\bar{d}_0} \bar{d}_0}{\sqrt{T}} \left(\log_+ \frac{\bar{d}_0}{d_\varepsilon}\right) \left(\log \frac{1}{\delta}\right) + \text{ERR}_N\left(\frac{T}{N}\right)\right),$$

which finishes the proof. $\square$

## C.5. Omitted Details of Base Algorithm

In this part, we introduce the base algorithm UNIXGRAD, with its convergence rate analysis.

**Lemma 5** (Base Algorithm with Bounded Domain). *Assuming that domain $\mathcal{X}$ is bounded by $D$, $\ell(\mathbf{x})$ is convex and $(L_\nu, \nu)$-Hölder smooth, and under Assumption 1, UNIXGRAD (Kavis et al., 2019), i.e. Algorithm 5, when given $D$, ensures that with probability at least $1 - \delta$:*

$$\ell(\overline{\mathbf{x}}_T) - \min_{\mathbf{x} \in \mathcal{X}} \ell(\mathbf{x}) \leq \mathcal{O}\left(\frac{L_\nu D^{1+\nu}}{T^{\frac{1+3\nu}{2}}} + \frac{\theta_{T,\delta} \Delta_D D}{\sqrt{T}}\right),$$

*where $\Delta_D$ is the maximum noise bound defined in Eq. (2), and $\theta_{T,\delta} \triangleq \log \frac{\log T}{\delta}$.*

*Proof of Lemma 5.* To begin with, we introduce the anytime online-to-batch conversion lemma (Cutkosky, 2019), which is a key ingredient for the analysis of UNIXGRAD.

**Lemma 6** (Cutkosky (2019)). *If the objective function $\ell(\cdot)$ is convex, then it holds that:*

$$\alpha_{1:T} \left[\ell(\overline{\mathbf{x}}_T) - \ell(\mathbf{x}^\star)\right] \leq \sum_{t=1}^{T} \langle \alpha_t \nabla \ell(\overline{\mathbf{x}}_t), \mathbf{x}_t - \mathbf{x}^\star \rangle, \tag{26}$$

*where $\overline{\mathbf{x}}_t \triangleq \frac{1}{\alpha_{1:t}} \sum_{s=1}^{t} \alpha_s \mathbf{x}_s$, $\alpha_t > 0$ for all $t \in [T]$, and $\alpha_{1:t} \triangleq \sum_{s=1}^{t} \alpha_s$.*

By Lemma 6 with $\alpha_t = t$, we have:

$$\alpha_{1:T} \left[\ell(\overline{\mathbf{x}}_T) - \ell(\mathbf{x}^\star)\right] \leq \sum_{t=1}^{T} \alpha_t \langle \nabla \ell(\overline{\mathbf{x}}_t), \mathbf{x}_t - \mathbf{x}^\star \rangle = \underbrace{\sum_{t=1}^{T} \alpha_t \langle \nabla \ell(\overline{\mathbf{x}}_t) - \mathbf{g}(\overline{\mathbf{x}}_t), \mathbf{x}_t - \mathbf{x}^\star \rangle}_{\text{GAP}_T} + \underbrace{\sum_{t=1}^{T} \alpha_t \langle \mathbf{g}(\overline{\mathbf{x}}_t), \mathbf{x}_t - \mathbf{x}^\star \rangle}_{\text{REG}_T}. \tag{27}$$

To bound $\text{GAP}_T$, we apply Lemma 9 with $X_t = \langle \nabla \ell(\overline{\mathbf{x}}_t) - \mathbf{g}(\overline{\mathbf{x}}_t), \mathbf{x}_t - \mathbf{x}^\star \rangle$, $\widehat{X}_t = 0$, and $c = \Delta_D D$ where $\Delta_D \triangleq \max_{\|\mathbf{x}\| \leq D} \Delta(\mathbf{x})$, then with probability at least $1 - \delta$, it holds that:

$$\text{GAP}_T \leq 8T \sqrt{\theta_{T,\delta} D^2 \sum_{t=1}^{T} \|\nabla \ell(\overline{\mathbf{x}}_t) - \mathbf{g}(\overline{\mathbf{x}}_t)\|^2 + \theta_{T,\delta}^2 \Delta_D^2 D^2} \leq \mathcal{O}\left(\sqrt{\theta_{T,\delta}} \Delta_D D T^{\frac{3}{2}} + \theta_{T,\delta} \Delta_D D T\right). \tag{28}$$

And for $\text{REG}_T$, following the analysis of Kavis et al. (2019, Theorem 4), i.e., starting from their Eq. (11):

$$\text{REG}_T \leq \frac{7D}{2} \sqrt{1 + \sum_{t=1}^{T} \alpha_t^2 \|\mathbf{g}(\overline{\mathbf{x}}_t) - \mathbf{g}(\widetilde{\mathbf{x}}_t)\|^2} - \frac{1}{2} \sum_{t=1}^{T} \frac{1}{\eta_{t+1}} \|\mathbf{x}_t - \widehat{\mathbf{x}}_t\|^2.$$

Next, we follow the same steps in Zhao et al. (2025, Theorem 2). By Lemma 11, for any $\beta > 0$, denote by $L = \beta^{\frac{\nu-1}{1+\nu}} L_\nu^{\frac{2}{1+\nu}}$:

$$\|\nabla \ell(\mathbf{x}) - \nabla \ell(\mathbf{y})\|^2 \leq L^2 \|\mathbf{x} - \mathbf{y}\|^2 + 4L\beta. \tag{29}$$

Let $A_t \triangleq \sum_{s=1}^{t} \alpha_s^2 \|\mathbf{g}(\overline{\mathbf{x}}_s) - \mathbf{g}(\widetilde{\mathbf{x}}_s)\|^2$. WLOG assume $\sqrt{A_T} \geq 2LD$, otherwise we will finish the proof trivially. Define $t_0 \in [T-1]$ that, if $\sqrt{A_1} > 2LD$, let $t_0 = 1$, otherwise let $t_0 = \min\{t \in [T-1], \sqrt{A_{t+1}} > 2LD\}$. Then we have $\sqrt{A_{t_0}} \leq 2LD$, while for all $t_0 + 1 \leq t \leq T$ it holds that $\sqrt{A_t} > 2LD$. Continuing with the inequality:

$$\text{REG}_T \leq \frac{7D}{2} \sqrt{1 + A_T} - \frac{1}{2} \sum_{t=1}^{T} \frac{1}{\eta_{t+1}} \|\mathbf{x}_t - \widehat{\mathbf{x}}_t\|^2$$

$$\leq \frac{7D}{2} \sqrt{1 + A_{t_0}} + \frac{7D}{2} \sqrt{\sum_{t=t_0+1}^{T} \alpha_t^2 \|\mathbf{g}(\overline{\mathbf{x}}_t) - \mathbf{g}(\widetilde{\mathbf{x}}_t)\|^2 - \frac{1}{2} \sum_{t=1}^{T} \frac{1}{\eta_{t+1}} \|\mathbf{x}_t - \widehat{\mathbf{x}}_t\|^2}$$

$$\leq \frac{7D}{2}(1 + 2LD) + \frac{7D}{2} \sqrt{\sum_{t=t_0+1}^{T} \alpha_t^2 \|\mathbf{g}(\overline{\mathbf{x}}_t) - \nabla \ell(\overline{\mathbf{x}}_t) + \nabla \ell(\overline{\mathbf{x}}_t) - \nabla \ell(\widetilde{\mathbf{x}}_t) + \nabla \ell(\widetilde{\mathbf{x}}_t) - \mathbf{g}(\widetilde{\mathbf{x}}_t)\|^2 - \frac{1}{2} \sum_{t=1}^{T} \frac{1}{\eta_{t+1}} \|\mathbf{x}_t - \widehat{\mathbf{x}}_t\|^2}$$

$$\leq \frac{7D}{2}(1 + 2LD) + \frac{7D}{2} \Delta_D T^{\frac{3}{2}} + \frac{7D}{2} \sqrt{\sum_{t=t_0+1}^{T} \alpha_t^2 \|\nabla \ell(\overline{\mathbf{x}}_t) - \nabla \ell(\widetilde{\mathbf{x}}_t)\|^2 - \frac{1}{2} \sum_{t=1}^{T} \frac{1}{\eta_{t+1}} \|\mathbf{x}_t - \widehat{\mathbf{x}}_t\|^2},$$

where we use the definition of $t_0$. Then we apply Eq. (29), with the definition $\alpha_{1:t} = \sum_{s=1}^{t} \alpha_s$,

$$\frac{7D}{2} \sqrt{\sum_{t=t_0+1}^{T} \alpha_t^2 \|\nabla \ell(\overline{\mathbf{x}}_t) - \nabla \ell(\widetilde{\mathbf{x}}_t)\|^2 - \frac{1}{2} \sum_{t=1}^{T} \frac{1}{\eta_{t+1}} \|\mathbf{x}_t - \widehat{\mathbf{x}}_t\|^2}$$

$$\overset{(29)}{\leq} \frac{7D}{2} \sqrt{\sum_{t=t_0+1}^{T} \alpha_t^2 L^2 \|\overline{\mathbf{x}}_t - \widetilde{\mathbf{x}}_t\|^2 + 4L\beta T^3 - \frac{1}{2} \sum_{t=1}^{T} \frac{1}{\eta_{t+1}} \|\mathbf{x}_t - \widehat{\mathbf{x}}_t\|^2}$$

$$\leq \frac{7D}{2} \sqrt{\sum_{t=t_0+1}^{T} \frac{\alpha_t^4 L^2}{\alpha_{1:t}^2} \|\mathbf{x}_t - \widehat{\mathbf{x}}_t\|^2} - \frac{1}{2} \sum_{t=1}^{T} \frac{1}{\eta_{t+1}} \|\mathbf{x}_t - \widehat{\mathbf{x}}_t\|^2 + 7D\sqrt{L\beta T^3},$$

where in the last line we use the definitions of $\overline{\mathbf{x}}_t$ and $\widetilde{\mathbf{x}}_t$. Since $\alpha_t = t$, we have $\frac{\alpha_t^4}{\alpha_{1:t}^2} = \frac{4t^4}{t^2(1+t)^2} \leq 4$, then

$$\frac{7D}{2} \sqrt{\sum_{t=t_0+1}^{T} \frac{\alpha_t^4 L^2}{\alpha_{1:t}^2} \|\mathbf{x}_t - \widehat{\mathbf{x}}_t\|^2} - \frac{1}{2} \sum_{t=1}^{T} \frac{1}{\eta_{t+1}} \|\mathbf{x}_t - \widehat{\mathbf{x}}_t\|^2$$

$$\leq 7D \sqrt{\sum_{t=t_0+1}^{T} L^2 \|\mathbf{x}_t - \widehat{\mathbf{x}}_t\|^2} - \frac{1}{2} \sum_{t=1}^{T} \frac{1}{\eta_{t+1}} \|\mathbf{x}_t - \widehat{\mathbf{x}}_t\|^2$$

$$\leq \frac{49LD^2}{2} + \frac{1}{2} \sum_{t=t_0+1}^{T} \left( L - \frac{1}{\eta_{t+1}} \right) \|\mathbf{x}_t - \widehat{\mathbf{x}}_t\|^2$$

$$= \frac{49LD^2}{2} + \frac{1}{2} \sum_{t=t_0+1}^{T} \left( L - \frac{\sqrt{1+A_t}}{2D} \right) \|\mathbf{x}_t - \widehat{\mathbf{x}}_t\|^2 \leq \frac{49LD^2}{2},$$

where in the second inequality we use $\sqrt{ab} \leq a/2 + b/2$ for $a, b > 0$, and in the last line we use the definition of $\eta_{t+1} = \frac{2D}{\sqrt{1+A_t}}$, and the definition of $t_0$ such that for all $t > t_0$, $\sqrt{A_t} \geq 2LD$. Combining everything together, we have

$$\text{REG}_T \leq \mathcal{O}\left( LD^2 + D\sqrt{L\beta T^3} + \Delta_D D T^{\frac{3}{2}} \right). \tag{30}$$

Setting $\beta = L_\nu D^{1+\nu} T^{\frac{-(3+3\nu)}{2}}$, substituting Eq. (28) and Eq. (30) into Eq. (27), with probability at least $1 - \delta$, it holds that:

$$\ell(\overline{\mathbf{x}}_T) - \ell(\mathbf{x}^\star) \leq \mathcal{O}\left( \frac{L_\nu D^{1+\nu}}{T^{\frac{1+3\nu}{2}}} + \frac{\theta_{T,\delta} \Delta_D D}{\sqrt{T}} \right),$$

which finishes the proof. $\qquad\square$

## D. Supporting Lemmas

In this section, we provide supporting lemmas for the main results.

**Lemma 7** (Lemma 9 of Attia & Koren (2024)). *Let $\ell : \mathcal{X} \to \mathbb{R}$ and $\widetilde{\ell}$ a zeroth-order oracle of $\ell$ such that for all $\mathbf{x} \in \mathcal{X}$, $\mathbb{E}[\widetilde{\ell}(\mathbf{x}) \mid \mathbf{x}] = \ell(\mathbf{x})$ and $|\widetilde{\ell}(\mathbf{x}) - \ell(\mathbf{x})| \leq \sigma_\ell$ with some $\sigma_\ell > 0$. Given candidates $\mathbf{x}_1, \cdots, \mathbf{x}_N$, let $\overline{\mathbf{x}} = \arg\min_{i \in [N]} \sum_{j=1}^{M} \widetilde{\ell}_j(\mathbf{x}_i)$, where $\widetilde{\ell}_j(\mathbf{x}_i)$ means the $j$-th independent function evaluation at $\mathbf{x}_i$. Then for any $\delta \in (0, 1)$, with probability at least $1 - \delta$:*

$$\ell(\overline{\mathbf{x}}) \leq \min_{i \in [N]} \frac{1}{M} \sum_{j=1}^{M} \widetilde{\ell}_j(\mathbf{x}_i) + \sqrt{\frac{2\sigma_\ell^2 \log \frac{2N}{\delta}}{M}} \leq \min_{i \in [N]} \ell(\mathbf{x}_i) + \sqrt{\frac{8\sigma_\ell^2 \log \frac{2N}{\delta}}{M}}.$$

**Lemma 8** (Sample Gradient). *Under Assumption 1, for any $\mathbf{x} \in \mathcal{X}$, independently sample $\mathbf{g}(\mathbf{x})$ for $M$ times, and define $\widehat{\mathbf{g}} = \frac{1}{M} \sum_{t=1}^{M} \mathbf{g}_t(\mathbf{x})$ where $\mathbf{g}_t(\mathbf{x})$ is the $t$-th gradient query on $\mathbf{x}$. Then for any $\delta \in (0, 1)$, with probability at least $1 - \delta$:*

$$\|\widehat{\mathbf{g}} - \nabla\ell(\mathbf{x})\| \leq \mathcal{O}\left( \frac{\Delta(\mathbf{x})\sqrt{\log_+ \frac{1}{\delta}}}{\sqrt{M}} \right).$$

*Proof of Lemma 8.* By Lemma 10, for any $\varepsilon > 0$:

$$\Pr\left[ \|\widehat{\mathbf{g}} - \nabla\ell(\mathbf{x})\| \geq \frac{\sqrt{2}(1+\varepsilon)\Delta(\mathbf{x})}{\sqrt{M}} \right] = \Pr\left[ \left\| \sum_{t=1}^{M} (\mathbf{g}_t(\mathbf{x}) - \nabla\ell(\mathbf{x})) \right\| \geq \sqrt{2}(1+\varepsilon)\Delta(\mathbf{x})\sqrt{M} \right] \leq \exp(-\varepsilon^2/3).$$

Solving $\delta = \exp(-\varepsilon^2/3)$, then with probability at least $1 - \delta$:

$$\|\widehat{\mathbf{g}} - \nabla\ell(\mathbf{x})\| \leq \mathcal{O}\left(\frac{\Delta(\mathbf{x})\sqrt{\log_+ \frac{1}{\delta}}}{\sqrt{M}}\right),$$

which completes the proof. □

**Lemma 9** (Lemma 7 of Ivgi et al. (2023))**.** *Let $\{\alpha_i\}_{i=1}^\infty$ be non-negative and non-decreasing sequence. Let $X_t$ be a martingale difference sequence adapted to $\mathcal{F}_t$ such that $|X_t| \leq c$ with constant $c > 0$. Then for all $\delta \in (0, 1)$, and $\widehat{X}_t \in \mathcal{F}_{t-1}, |\widehat{X}_t| \leq c$, it holds that*

$$\Pr\left[\exists t \in [T] : \left|\sum_{i=1}^t \alpha_i X_i\right| \geq 8\alpha_t\sqrt{\theta_{t,\delta}\sum_{i=1}^t \left(X_i - \widehat{X}_i\right)^2 + c^2\theta_{t,\delta}^2}\right] \leq \delta, \tag{31}$$

*where $\theta_{t,\delta} \triangleq \log \frac{60\log(6t)}{\delta}$.*

**Lemma 10** (Lemma 2.3 of Ghadimi & Lan (2013))**.** *Let $X_1, \ldots, X_n \in \mathbb{R}^d$ be a martingale difference sequence with respect to $\mathcal{F}_1, \ldots, \mathcal{F}_n$. Assuming $\mathbb{E}[\exp(\frac{\|X_i\|^2}{\sigma^2})|\mathcal{F}_1, \ldots, \mathcal{F}_{i-1}] \leq e$, for any $\varepsilon > 0$:*

$$\Pr\left[\left\|\sum_{i=1}^n X_i\right\| \geq \sqrt{2}(1 + \varepsilon)\sigma\sqrt{n}\right] \leq \exp(-\varepsilon^2/3). \tag{32}$$

The following lemma is a direct combination of Nesterov (2015, Lemma 1) and Devolder et al. (2014, Theorem 1).

**Lemma 11.** *Suppose the function $f$ is $(L_\nu, \nu)$-Hölder smooth. Then, for any $\delta > 0$, denoting by $L = \delta^{\frac{\nu-1}{1+\nu}}L_\nu^{\frac{2}{1+\nu}}$, it holds that for all $\mathbf{x}, \mathbf{y} \in \mathbb{R}^d$:*

$$\|\nabla f(\mathbf{x}) - \nabla f(\mathbf{y})\|^2 \leq L^2\|\mathbf{x} - \mathbf{y}\|^2 + 4L\delta. \tag{33}$$

# E. Omitted Experimental Results for Section 6

We provide the experimental results of the convex setting in Table 4 and Table 5 for Algorithm 3 GRASP-C with two options, OPTION-I and OPTION-II, whose setup and measure are described in Section 6. Both OPTION-I and OPTION-II show similar performance, as the different options only influence the upper bound of the search range, and their best candidates consistently use the same hyper-parameters. In both tables, small values of $\rho$ (between 0.08 and 0.16) also indicate that our method achieves competitive performance with the optimally tuned one. Moreover, the performance is not sensitive to the initial sampling budget $M$, again suggesting that we can use a smaller $M$ in practice.

*Table 4.* GRASP-C OPTION-I. We vary inputs $(d_\varepsilon, L_\varepsilon)$ in $\{0.001, 0.01, 0.1\} \times \{0.001, 0.01, 0.1\}$, and the initial sampling budget $M$ from $T/4$ to $T/4096$. We report the relative difference in function value $\rho$ between our method and the optimally tuned one.

| Inputs | | $\rho$ | | | | | |
|---|---|---|---|---|---|---|---|
| $d_\varepsilon$ | $L_\varepsilon$ | $M = T/4$ | $M = T/16$ | $M = T/64$ | $M = T/256$ | $M = T/1024$ | $M = T/4096$ |
| 0.001 | 0.001 | 0.1640 | 0.1419 | 0.1419 | 0.1419 | 0.1419 | 0.1419 |
| 0.001 | 0.01 | 0.1631 | 0.1419 | 0.1419 | 0.1419 | 0.1419 | 0.1419 |
| 0.001 | 0.1 | 0.1610 | 0.1419 | 0.1419 | 0.1408 | 0.1408 | 0.1411 |
| 0.01 | 0.001 | 0.1303 | 0.1210 | 0.1190 | 0.1190 | 0.1190 | 0.1190 |
| 0.01 | 0.01 | 0.1303 | 0.1197 | 0.1190 | 0.1190 | 0.1190 | 0.1190 |
| 0.01 | 0.1 | 0.1303 | 0.1197 | 0.1190 | 0.1190 | 0.1187 | 0.1187 |
| 0.1 | 0.001 | 0.0920 | 0.0873 | 0.0873 | 0.0873 | 0.0873 | 0.0873 |
| 0.1 | 0.01 | 0.0920 | 0.0873 | 0.0873 | 0.0866 | 0.0866 | 0.0866 |
| 0.1 | 0.1 | 0.0898 | 0.0873 | 0.0866 | 0.0866 | 0.0866 | 0.0866 |

*Table 5.* GRASP-C OPTION-II. We set $\ell_\varepsilon^\star = 0$, and vary inputs $(d_\varepsilon, L_\varepsilon)$ in $\{0.001, 0.01, 0.1\} \times \{0.001, 0.01, 0.1\}$, and the initial sampling budget $M$ from $T/4$ to $T/4096$. We report the relative difference in function value $\rho$ between our method and the optimally tuned one.

| Inputs | | $\rho$ | | | | | |
|---|---|---|---|---|---|---|---|
| $d_\varepsilon$ | $L_\varepsilon$ | $M = T/4$ | $M = T/16$ | $M = T/64$ | $M = T/256$ | $M = T/1024$ | $M = T/4096$ |
| 0.001 | 0.001 | 0.1640 | 0.1419 | 0.1419 | 0.1419 | 0.1419 | 0.1419 |
| 0.001 | 0.01 | 0.1631 | 0.1419 | 0.1419 | 0.1419 | 0.1419 | 0.1419 |
| 0.001 | 0.1 | 0.1616 | 0.1419 | 0.1419 | 0.1411 | 0.1411 | 0.1411 |
| 0.01 | 0.001 | 0.1303 | 0.1210 | 0.1190 | 0.1190 | 0.1190 | 0.1190 |
| 0.01 | 0.01 | 0.1303 | 0.1200 | 0.1190 | 0.1190 | 0.1190 | 0.1190 |
| 0.01 | 0.1 | 0.1303 | 0.1197 | 0.1190 | 0.1190 | 0.1187 | 0.1187 |
| 0.1 | 0.001 | 0.0920 | 0.0873 | 0.0873 | 0.0873 | 0.0873 | 0.0873 |
| 0.1 | 0.01 | 0.0920 | 0.0873 | 0.0873 | 0.0866 | 0.0866 | 0.0866 |
| 0.1 | 0.1 | 0.0898 | 0.0873 | 0.0866 | 0.0866 | 0.0866 | 0.0866 |

