# OpenReview forum: "Towards Fully Parameter-Free Stochastic Optimization: Grid Search with Self-Bounding Analysis"
_ICML.cc/2026/Conference — ICML 2026 regular_

### Official Review · Reviewer_ckSH · 2026-02-17

**Soundness:** 4
**Presentation:** 4
**Significance:** 3
**Originality:** 3
**Overall Recommendation:** 6
**Confidence:** 4

**Summary:**

The paper proposes a generic procedure for determining the parameter search range: it suffices to set the upper bound range to values in which the base algorithm’s theoretical guarantees just becoming vacuous. This generic methods works for any base algorithm requires problem dependent hyper-parameter tunings. As a result, this provides an effective grid search ranging for Attia & Koren (2024) and yielded fully parameter-free algorithms for smooth stochastic optimization that achieve optimal rates in both convex and non-convex settings . In the convex case, this naturally further extends to Holder-smooth objectives under a mild lower-boundedness assumption on the loss function.

**Compliance With Llm Reviewing Policy:**

Affirmed.

**Key Questions For Authors:**

For the convex holder smooth case, I wonder why the value of $\ell (x^*)$
cannot be guessed using the proposed grid search method?


It seems that negative values requires more attention, One can set $\ell_\epsilon < 0 $ as the initial guess value as the lower bound to $\ell(x^*)$,

so $p_i = - \ell(x^*), p^i_{\epsilon} = -\ell_\epsilon$ in the notations of eqn 4 & 5 will still fit in the self bounding analysis framework together with adaptation to $d_0$

**Limitations:**

Yes

**Strengths And Weaknesses:**

Strength:
- the paper is well written, and clearly communicated the core idea of self bounding as a generic method which might have future applications in other avenue.
- the paper achieved claimed ``fully parameter-free'' in smooth stochastic optimization for both convex and non-convex case. For the convex case, the result also extend to holder smoothness if function is lower bounded, or without this information for a slightly worse result.
- the achieved results were fairly compared with existing results in terms of problem dependent parameters, convergence rates in terms of leading terms and other auxiliary terms. For the leading terms, the result from this paper are always tight, for auxiliary terms, the paper had extensive discussion on how it might fail short or exceed. For example the ensemble error when comparing with Kreisler et al. (2024) and Attia & Koren (2024). Nevertheless, the presented work is fully parameter-free.

Weakness:
- nothing in particular

---

> ### Author Rebuttal · Authors · 2026-03-31
>
> Thank you for your strong support for our paper! We answer your question in the following.
>
> ---
>
> **Q.** For the convex holder smooth case, I wonder why the value of $\ell(x^*)$ cannot be guessed using the proposed grid search method?
>
> **A.** Thanks for the insightful question! At present, we do not yet know how to incorporate $\ell(x^ *)$ into grid search. One direct reason is that, under our current method, we require $\epsilon^{\text{TAR}}(T,p^1,\dots,p^m)$ to be a *non-decreasing* function of the $p_i$'s (Line 207 after Eq. 4), so setting $p_i=-\ell(x^ *)$ does not satisfy this requirement. However, we believe this is essentially because $\ell(x^ *)$ is a relatively special parameter, unlike most problem parameters that appear in the best-tuned convergence rate. Precisely because $\ell(x^ *)$ is special, in many practical scenarios its lower bound is generally easier to obtain (for example, non-negative loss functions, such as the commonly used squared loss, cross-entropy loss, and hinge loss). As a conjecture, prior knowledge of $l_\epsilon^ *$ (or more generally, the suboptimality gap) may involve a lower bound when aiming for universality, especially considering that an extension of Polyak step-size methods for universality also requires $l_\epsilon^ *$ [1], and recent progress in convex universal optimization requires a pre-specified accuracy $\epsilon$ [2].
>
> [1] Revisiting the Polyak step size, ArXiv:1905.00313
>
> [2] A simple uniformly optimal method without line search for convex optimization, MP 2025
>
> ---
>
> We believe that our paper makes a step forward in the field of parameter-free stochastic optimization.
> Once again, we appreciate your positive feedback and valuable encouragement. We would be happy to provide any further clarifications needed.

---

> > ### Author Rebuttal · Reviewer_ckSH · 2026-04-03
> >
> > thank you for the clarification, i maintain the evaluation.

---

> > > ### Author Response · Authors · 2026-04-06
> > >
> > > Thank you again for your strong support for our paper and insightful question!

---

### Official Review · Reviewer_dEJt · 2026-03-12

**Soundness:** 3
**Presentation:** 3
**Significance:** 3
**Originality:** 3
**Overall Recommendation:** 5
**Confidence:** 2

**Summary:**

This paper proposes grid search techniques integrated with self-bounding analysis to realize fully parameter-free stochastic optimization. In doing so, it clearly defines the problem by distinctly separating the concepts of 'fully parameter-free' and 'partially parameter-free'. The authors develop separate algorithms, GRASP-NC and GRASP-C, for convex and non-convex cases respectively, and derive the theoretical convergence rates for each. Furthermore, the paper highlights the novelty and strengths of its approach through a detailed comparison with a state-of-the-art algorithm, U-DOG.

**Compliance With Llm Reviewing Policy:**

Affirmed.

**Final Justification:**

I appreciate the authors' rebuttals. The additional experiments clarify the paper's contributions, and I have no further concerns. I also read all other reviews and now I believe this work gives a strong contribution. I am happy to raise my score to 5, with lowering my confidence to 2.

**Key Questions For Authors:**

- Are there any plans to include empirical evaluations (e.g., code implementation and experiments) of GRASP-C and GRASP-NC to demonstrate their practical viability?
- Do the authors consider the strictly bounded noise assumption to be sufficiently general for real-world applications?
- How sensitive are the algorithms to reducing the initial sampling budgets ($T/4$ and $T/8$)? If the sensitivity is low, can these large overheads be further reduced in practice?

**Limitations:**

The author suggested two clear directions for future work.

**Strengths And Weaknesses:**

### Strengths
- The integration of self-bounding analysis with grid search to achieve fully parameter-free stochastic optimization is highly novel. Successfully removing the dependence on problem-specific constants (such as d_0) while only incurring a minimal logarithmic penalty is a significant theoretical breakthrough.
- The paper makes a crucial contribution by explicitly defining the boundary between "fully" and "partially" parameter-free settings, which establishes a new standard for evaluating future parameter-free algorithms in stochastic optimization.
- The paper is very well-structured and easy to follow. By explaining the core idea in a simple deterministic setting first, and then moving to complex stochastic settings for both non-convex and convex case. The authors make the difficult mathematical proofs much easier to understand.

---

### Weaknesses
- The paper completely lacks empirical experiments, which is a critical weakness that severely undermines its claims. Given the practical nature of the proposed grid search and ensemble mechanisms, empirical validation is absolutely essential to prove the framework's actual feasibility and persuasiveness.
- In GRASP-NC, choosing a small L_epsilon leads to an extremely large maximum step size. Running candidates with such massive step sizes in practice will inevitably cause numerical instability and divergence, yet no practical safeguards are discussed.
- The framework relies on the 'almost-surely bounded noise' assumption, limiting its generality. Unlike U-DOG (Kreisler et al., 2024), which extends its proofs to more realistic sub-Gaussian noise, this paper fails to move beyond this restrictive condition.

---

> ### Author Rebuttal · Authors · 2026-03-31
>
> Thanks for recognizing our theoretical contributions! We acknowledge that empirical validation is essential to demonstrate the framework's actual feasibility and persuasiveness, and we conduct experiments in response to your suggestion.
>
> We consider both convex and non-convex settings. Here we only present partial results in the non-convex case, and other results also support our claims. Complete results can be found in [anonymous-link](https://anonymous.4open.science/r/ICML26-215Fd1/experiment.pdf).
>
> - **Setup.** We use ResNet-18 with an added linear layer, CIFAR-10, cross-entropy loss, with total budget $T=10^4$ for initial sampling and training.
> - **Measure.** To compare with the optimally tuned one, we define **relative difference $\rho$**. Marking our method's output as $x^{out}$. We perform a more fine-grained search to find the best-tuned hyper-parameters, allocate the *entire budget* to it, and get $x^{tuned}$. Define $\rho=\frac{\\|\nabla \ell(x^{out})\\| - \\| \nabla \ell(x^{tuned})\\|}{\\|\nabla \ell(x^{tuned})\\|}$. Smaller $\rho$ indicates better performance.
> - **Algorithmic Inputs.** We set $\delta = 0.05$ and test each pair $(L _\epsilon,F _\epsilon)\in\\{0.001,0.01,0.1\\}\times\\{0.001, 0.01, 0.1\\}$.
> - **Sensitivity Study.** We also investigate $T/M\in\\{4, 16, 64, 256, 1024, 4096\\}$, where $M$ is the budget for initial sampling.
>
> ---
>
> **Q1.** Are there any plans to include empirical evaluations (e.g., code implementation and experiments) of GRASP-C and GRASP-NC to demonstrate their practical viability?
>
> **A1.** We vary the inputs, and the results show that our method, *while being parameter-free*, obtains **competitive convergence with optimally tuned methods**. In Table 1 below, $\rho$ lies between 0.11~0.23, meaning the actual impact of the log factor in theory only results in a multiplier of 1.11 to 1.23, justifying the claim of competitiveness.
>
> Table 1: GRASP-NC with $M=T/4$.
>
> $L_\epsilon$|$F_\epsilon$|$\rho$
> :-:|:-:|:-:
> 0.001|0.001|0.1836
> 0.001|0.01|0.1468
> 0.001|0.1|0.1468
> 0.01|0.001|0.2336
> 0.01|0.01|0.2094
> 0.01|0.1|0.2094
> 0.1|0.001|0.2264
> 0.1|0.01|0.2023
> 0.1|0.1|0.1141
>
> ---
>
> **Q2.** How sensitive are the algorithms to reducing the initial sampling budgets ($T/4$ and $T/8$)? If the sensitivity is low, can these large overheads be further reduced in practice?
>
> **A2.** Reducing the initial sample budget has *little* impact, and may *improve* the results thanks to the saved budget. In Table 2 below, as $M$ decreases, $\rho$ does not get larger. This phenomenon can also be explained below: less initial sampling offers more budget for the later grid search, exhibiting a trade-off between initial exploration and latter exploitation. Therefore, sampling less initially might not necessarily hurt the performance.
>
> Table 2: GRASP-NC with $L _\epsilon=0.001,F _\epsilon=0.001$.
>
> $T/M$|$\rho$
> :-:|:-:
> 4|0.1836
> 16|0.1699
> 64|0.1699
> 256|0.1699
> 1024|0.1699
> 4096|0.1699
> ---
> **Q3.** In GRASP-NC, choosing a small L_epsilon leads to an extremely large maximum step size. Running candidates with such massive step sizes in practice will inevitably cause numerical instability and divergence, yet no practical safeguards are discussed.
>
> **A3.** Table 3 below shows that our method remains competitive with different scales of $L_\epsilon$. In practice, the smoothness parameter is typically not too small, e.g., where the smoothness scales with the gradient norm [1]. Therefore, it is not suggested to pick an extremely small $L_\epsilon$. We will discuss this point further in the revised version.
>
> [1] Why gradient clipping accelerates training: A theoretical justification for adaptivity
>
> Table 3: GRASP-NC with $F_\epsilon=0.01,M=T/4$.
>
> $L_\epsilon$|$\rho$
> :-:|:-:
> 0.001|0.1468
> 0.01|0.2094
> 0.1|0.2023
>
>
> We will incorporate this experimental section into the next version of the paper. Thanks for your remarkable suggestion!
>
> ---
>
> **Q4.** Do the authors consider the strictly bounded noise assumption to be sufficiently general for real-world applications?
>
> **A4.** Thanks for the question. Our work currently uses but is not stricted to the common bounded noise assumption. It can be extended to more relaxed noise models, such as sub-Gaussian, where a reduction can transform sub-Gaussian noises into a bounded setting, as Kreisler et al. did. Moreover, our method remains free from noise variance. We will add a remark regarding the extension to sub-Gaussian in the revised version.
>
> ---
> We believe that our work, through the highly novel self-bounding analysis, makes valuable contributions to the community. If our responses have adequately addressed your concerns, we kindly request a reevaluation of our paper's score. We are happy to provide further clarifications if needed during the author-reviewer discussions.

---

> > ### Author Rebuttal · Reviewer_dEJt · 2026-04-04
> >
> > I appreciate the authors' rebuttals. The additional experiments clarify the paper's contributions, and I have no further concerns. I also read all other reviews and now I believe this work gives a strong contribution. I am happy to raise my score to 5, with lowering my confidence to 2.

---

> > > ### Author Response · Authors · 2026-04-06
> > >
> > > Thank you for your further affirmation! We will incorporate this experimental section into the next version of the paper. Thanks again for your valuable and helpful review!

---

### Official Review · Reviewer_u74K · 2026-03-12

**Soundness:** 2
**Presentation:** 3
**Significance:** 2
**Originality:** 2
**Overall Recommendation:** 3
**Confidence:** 4

**Summary:**

The paper introduces GRASP, a grid search framework for fully parameter-free stochastic optimization under convex and nonconvex settings. This approach proposes a self-bounding analysis technique. Meanwhile, the method analytically derives a bounded search space for algorithmic parameters. The authors establish high-probability convergence guarantees for both non-convex and convex settings. Additionally, the paper provides a refined model ensemble guarantee under an interpolated variance characterization.

**Compliance With Llm Reviewing Policy:**

Affirmed.

**Final Justification:**

I maintain the score of Weak Accept. The reasons are provided in acknowledgement.

**Key Questions For Authors:**

(1) Lemma 1 relies on the assumption in Equation (13)
> $VAR[\tilde{l}(x)|x] \leq V_0 + V_1(l(x) - l(x^*)).$

Could this assumption hold in nonconvex landscapes? If $V_1$ scales nonlinearly, would the theoretical guarantee become substantially weaker?

(2) In line 6 of Algorithm 2, the budget allocated to each base SGD run is $\lfloor T/2N \rfloor$. As $N$ increases, does the theoretical rate in Theorem 2 hide large constant concerning $N$?

(3) Could you provide explanations for the third and the fourth points in **Weaknesses** section

**Limitations:**

The primary limitation lies in the practical efficiency of the framework, since the algorithm divides the total oracle budget $T$ for multiple parallel runs and the initial evaluations.

In the theoretical analysis, these additional overheads are calculated as logarithmic terms. However, splitting the budget may reduce the performance of each single execution path.

**Strengths And Weaknesses:**

# Strengths
- The paper formally distinguishes between partially and fully parameter-free algorithms.
- The core idea is using a benchmark rate to upper-bound the search space of unknown parameters. This is theoretically sound and elegant.
- The theoretical results are solid, which provides near-optimal rates for non-convex optimization and demonstrates universality to Holder smoothness in the convex setting.
# Weaknesses
- The algorithms require substantial initial sampling merely to construct the grid. For example, GRASP-NC requires $T/4$ gradient oracle calls at $x^0$ , and GRASP-C requires $T/8$ gradient oracle calls at $x^0$. Although this is mainly for theoretical tightness, this seems impractical for real-world scenarios **with limited budgets**.
- The framework introduces the zeroth-order variance term $\sigma_l$ into the ensemble error, which would dominate the standard gradient variance bounds in noisy environments.
- For the universal convex setting OPTION-II, the algorithm requires a lower bound of the objective value $l_\epsilon^*$. If the value is strictly greater than $0$ and unknown, relying on such prior knowledge seems **somewhat inconsistent with the parameter-free claim** made in the title of the paper.
- Regarding $ERR_N(\cdot)$, you mention that it is to be specified in Section 5.1, but there does not seem to be a precise mathematical definition there. Since this quantity is important to your theoretical analysis, it should still be stated explicitly.

---

> ### Author Rebuttal · Authors · 2026-03-31
>
> Thanks for the feedback. Below we answer your questions and clarify our contributions, hoping to address your concerns.
>
> ---
>
> **Q1.** Could the assumption of $VAR[\tilde{l}(x)|x] \leq V_0 + V_1(l(x) - l(x^*))$ hold in nonconvex landscapes? If $V_1$ scales nonlinearly, would the theoretical guarantee become substantially weaker?
>
> **A1.** We use this assumption only in the convex case, as in the non-convex case we do not rely on the zeroth-order oracle. Canonical examples satisfying this assumption include over-parameterized neural networks and over-parameterized linear models [1,2,3].
>
> And it always holds that $V_0=\sigma_\ell^2$ and $V_1=0$, so our result strictly recovers the original one in [4]. This sharper guarantee for the ensemble is one of our contributions.
>
> [1] Aiming towards the minimizers: Fast convergence of SGD for overparametrized problems, NeurIPS 2023
>
> [2] The power of interpolation: Understanding the effectiveness of SGD in modern over-parametrized learning, ICML 2018
>
> [3] From continual learning to SGD and back: Better rates for continual linear models, arXiv:2504.04579
>
> [4] How Free is Parameter-Free Stochastic Optimization?, ICML 2024
>
> ---
>
> **Q2.** In line 6 of Algo 2, the budget allocated to each base SGD run is $\lfloor T/2N \rfloor$. As $N$ increases, does the theoretical rate in Thm 2 hide large constant of $N$?
>
> **A2.** Thanks for the question. We did **not** hide the dependence on $N$ in Thm 2. $N$ is $O(\text{poly}\log\frac{T\\|\hat{g}^0\\|}{L_\epsilon F_\epsilon})$, which has already been considered in Thm 2. Furthermore, this logarithmic factor, seen as the cost of achieving parameter-freeness, is usually unavoidable [4,5].
>
> [5] DoG is SGD's Best Friend: A Parameter-Free Dynamic Step Size Schedule, ICML 2023
>
> ---
>
> **Q3.** The method with OPTION-II requires a lower bound of the objective value $l_\epsilon^*$. If the value is strictly greater than $0$ and unknown, relying on such prior knowledge seems somewhat inconsistent with the parameter-free claim made in the title of the paper.
>
> **A3.** Thanks for the comment. We have explicitly marked the requirement of $l_\epsilon^{* }$ in Table 1(b) to avoid misleading the readers, and this method achieves **strong universality** under **Holder smoothness**. Additionally, in Line 143-146, we emphasize the key problem parameters that we primarily focus on, and do not highlight $l_\epsilon^*$ because we believe this parameter is generally easier to obtain (for example, non-negative loss functions, such as the commonly used squared loss, cross-entropy loss, and hinge loss mentioned in Line 352-353).
>
> ---
>
> **Q4.** Regarding $ERR_N(\cdot)$, you mention that it is to be specified in Section 5.1, but there does not seem to be a precise mathematical definition there. Since this quantity is important to your theoretical analysis, it should still be stated explicitly.
>
> **A4.** Thanks for the suggestion! We have provided the definition in Table 2, i.e., "$\ell(x^\text{out})-\ell(x^* )\leq O(\min_i\ell(x^i)-\ell(x^*)+Err_N(M))$, where $N$ candidates are sampled $M$ times, respectively." We will add an inline equation to state the definition formally.
>
> ---
>
> **Q5.** The algorithms require substantial initial sampling merely to construct the grid: GRASP-NC and GRASP-C need $T/4$ and $T/8$ gradient oracle calls at $x^0$. While it is mainly for theoretical tightness, this seems impractical for real-world scenarios with limited budgets.
>
> **A5.** Thanks for the comments. We conduct experiments on multiple choices of the initial sampling budget and show that **reducing the initial sampling budget does not hurt the empirical performance**, and may even improve results by saving budget for the optimization phase. More experimental details are available at [anonymous-link](https://anonymous.4open.science/r/ICML26-215Fd1/experiment.pdf).
> This phenomenon can also be explained as: less initial sampling offers more budget for the later grid search, exhibiting a trade-off between initial exploration and later exploitation. Therefore, sampling less initially might not necessarily hurt the performance.
>
> ---
>
> **Q6.** The work introduces the zeroth-order variance $\sigma_l$ in the ensemble error, which would dominate the standard gradient variance bounds in noisy environments.
>
> **A6.** Thanks for the sharp observation! We acknowledge that the $\sigma_l$ term is an important factor to consider, and it also appears in prior work [4]. Compared with [4], we make progress by showing that: under a more refined characterization of $VAR[\tilde{l}(x)|x] \leq V_0 + V_1(l(x) - l(x^*))$, this term can be improved, which is one of our contributions for the ensemble mechanism.
>
> ---
>
> We believe that our work makes valuable contributions to parameter-free optimization. If our responses have adequately addressed your concerns, we kindly request a reevaluation of our paper's score. We are happy to provide further clarifications if needed during the author-reviewer discussions.

---

> > ### Author Rebuttal · Reviewer_u74K · 2026-04-02
> >
> > Thanks for your detailed response. Unfortunately, my primary concern remains unresolved.
> >
> > 1. The algorithm requires $T/4$ or $T/8$ oracle calls during initialization to estimate $||\hat{g}^0||$. In Rebuttal A5, the authors argue that reducing the initial sampling budget does not harm performance and may even improve results. However, this defense lacks validity in a theoretical context.
> > If the budget is reduced, the grid search bounds $L_{max}$, $F_{max}$ and $\Delta_{max}$ lose their effectiveness. As a result, the algorithm becomes theoretically inefficient in practice. However, it is a method without strong theoretical guarantees in the experiments. This gap between theory and practice greatly weakens the contribution of the paper.
> >
> > 2. In Rebuttal A2, the authors argue that the number of base classifiers $N$ is only a logarithmic term $\mathcal{O}(\text{poly log} \dots)$, and thus theoretically acceptable. However, this argument overlooks convergence within a finite horizon in random optimization. In Algorithm 2, the budget per SGD instance is limited to $\lfloor T/(2N) \rfloor$. In Algorithm 3, it is further reduced to $\lfloor T/(2i(1+\ln N)) \rfloor$. This results in a severe compression of the actual optimization steps under a finite budget $T$. While $N$ is logarithmic asymptotically, splitting the limited $T$ into many short parallel trajectories prevents each base algorithm from fully converging.
> >
> > 3. In Rebuttal A6, the authors argue that they have refined the bound to $VAR \le V_0 + V_1(l(x) - l(x^*))$ to address the issue of zero-order variance $\sigma_l$. However, this approach still contains a constant term $V_0 = \sigma_l^2$. The robustness of traditional tuned SGD largely arises from its avoidance of querying zero-order function values. In high-noise environments, such as those with large batch differences or high randomness in data augmentation, the variance $\sigma_l$ of function evaluations can be large.
> >
> > In summary, I maintain my score. Thank you for your response, and I wish you all the best in improving the work.

---

> > > ### Author Response · Authors · 2026-04-05
> > >
> > > Below, we address your concerns in detail and clarify the paper's contributions, hoping to resolve your concerns.
> > >
> > > ---
> > >
> > > **Q7.** The method uses $T/4$ or $T/8$ oracle calls to estimate $\\|\hat{g}^0\\|$. In Rebuttal A5, the authors argue reducing the initial sampling budget does not harm performance and may even improve results. However, this defense lacks validity in a theoretical context. If the budget is reduced, the grid search bounds $L_\max$, $F_\max$ and $\Delta_\max$ lose their effectiveness. ...
> > >
> > > **A7.** We offer intuitions on the impact of reducing initial sampling budget:
> > >
> > > 1) Reducing the initial sampling does **not** have a significant impact in practice. To see this, in the proof, only if $x^0$ becomes the best candidate, the rate introduces an estimation error for $\\|\hat{g}^0\\|$, necessitating a sufficiently large sample size. However, in practice, the probability of "$x^0$ is the best candidate" is small, so reducing the initial sampling will not cause a substantial effect. Therefore, we **disagree** with "If the budget is reduced, the grid search bounds $L_\max$, $F_\max$ and $\Delta_\max$ lose their effectiveness".
> > > 2) There exists a trade-off in sampling less. Reducing the sample size increases the variance of $\\|\hat{g}^0\\|$, affecting the search range, but only increases the base learner number logarithmically (e.g., just 3-4 more learners in practice). On the other hand, it provides a larger total budget to the base learners, potentially allocating more to each and improving performance.
> > >
> > > ---
> > >
> > > **Q8.** In Rebuttal A2, the authors argue the number of base classifiers $N$ is only $O(\text{poly log}\dots)$, and thus theoretically acceptable. However, this argument overlooks convergence within a finite horizon in random optimization. In Algorithm 2, the budget per SGD instance is limited to $\lfloor T/(2N) \rfloor$. In Algorithm 3, it is further reduced to $\lfloor T/(2i(1+\ln N)) \rfloor$. This results in a severe compression of the actual optimization steps under a finite budget $T$. While $N$ is logarithmic asymptotically, splitting the limited $T$ into many short parallel trajectories prevents each base algorithm from fully converging.
> > >
> > > **A8.** Our A2 fully answers Q2 on if using $\lfloor T/2N \rfloor$ will import a factor of $N$ *theoretically*. Then the reviewer unexpectedly shifted their focus to the *practical* impact of splitting the budget, which to some extent conflates the boundaries between theory and practice. Our response is:
> > >
> > > 1. The theoretical discussion is relatively independent, and the analysis of convergence rate order is not related to the "finite horizon."
> > > 2. Our method offers a competitive theoretical convergence rate while being more parameter-free. Therefore, in our view, it is not appropriate to assess our paper solely from a purely empirical standpoint without accounting for these technical contributions. Convergence proofs from parameter-free optimization provide guarantees that heuristic methods cannot, and together they contribute to the development of optimization algorithms that are both principled and empirically effective.
> > >
> > > ---
> > >
> > > **Q9.** In Rebuttal A6, the authors argue they have refined the bound to $VAR \le V_0 + V_1(l(x) - l(x^*))$ to address the issue of zero-order variance $\sigma_l$. However, this approach still contains a constant term $V_0 = \sigma_l^2$. ..., the variance $\sigma_l$ of function evaluations can be large.
> > >
> > > **A9.** We further explain point by point.
> > >
> > > 1. The zeroth-order variance term can be adaptively improved via this more refined assumption. It is a **"problem-dependent"** characterization, where $V_0 = \sigma_l^2$ only holds in the *worst case* and is smaller than $\sigma_l^2$ otherwise. Moreover, it holds in the over-parameterized setting [1,2,3], where optimal points can nearly fit all samples (i.e., interpolation), implying that the variance at the optimum, $V_0$, is very small. Therefore, we improve the existing ensemble error of $\sqrt{\sigma_l N/T}$ to $\sqrt{V_0 N/T} + (\sigma_l + V_1)N/T$ with $V_0 \le \sigma_l$ and a lower order term of $O(1/T)$.
> > > 2. We only use this in the convex case. The SGD method you mentioned applies to the non-convex setting, whereas our non-convex method has not used zeroth-order variance at any point.
> > >
> > > ---
> > >
> > > Finally, we would like to clarify that the primary focus of this paper is to advance the **theoretical foundations of parameter-free optimization**. Specifically, to better understand its theoretical limits and to establish a **principled algorithmic framework** with the desired convergence guarantees.
> > >
> > > Accordingly, many design choices are made to preserve **theoretical rigor**. We also provide detailed explanations on how these designs can be further improved for practical applications. We appreciate the reviewer's careful attention to practical considerations, and we hope the reviewer will also account for and value the paper's **theoretical contributions** when evaluating our work.

---

### Official Review · Reviewer_1teK · 2026-03-12

**Soundness:** 4
**Presentation:** 4
**Significance:** 3
**Originality:** 4
**Overall Recommendation:** 5
**Confidence:** 4

**Summary:**

The paper proposes a grid-search meta-framework, GRASP, for stochastic optimization. Its main idea is a self-bounding analysis: instead of requiring known upper/lower bounds on unknown problem parameters, it derives an effective search range by comparing the target convergence rate to a trivial benchmark based on the initial point. The framework is instantiated into nonconvex smooth stochastic optimization, giving a fully parameter-free SGD-style method; convex smooth stochastic optimization; convex Hölder-smooth optimization.

**Compliance With Llm Reviewing Policy:**

Affirmed.

**Final Justification:**

I am happy/satisfied with the authors' rebuttal. Hence, I keep my positive assessment.

**Key Questions For Authors:**

Please see weakness above.

**Limitations:**

The major limitation, IMHO, is the lack of experiments, which restricts the assessment of the practical usefulness of the work, despite that it reads well theoretically.

**Strengths And Weaknesses:**

Strength:

1. Clear conceptual distinction between partial and full parameter-freeness. This is one of the paper’s most valuable contributions.
2. The self-bounding + grid-search template is flexible and reuses base algorithms rather than designing a new optimizer from scratch, which is quite valuable, IMHO.
3. the coverage of different categories of numerical optimization is nice: the paper addresses nonconvex, convex smooth, and convex Hölder-smooth classes.
4. Comparisons to prior work are well-motivated, especially versus Attia–Koren and Kreisler, which makes the paper strong.
5. I only managed to check the proofs of Thm. 3 and 4, which look good and rigorous.

Weakness:
1. The novelty is good but not very strong. The framework leans heavily on known ingredients: SGD bounds, UNIXGRAD, grid search, and prior ensemble ideas. The new part is mostly the self-bounding wrapper.
2. No experiments are provided. For a grid-search framework motivated partly by practice, this is a major omission. I am fully aware that this is a theory paper, yet, simultaneously, it proposes a new grid-search method. It would be good to see how it works in some small examples/algorithms/problems.
3. The practical oracle overhead is unclear. For example, Algorithm 2 spends $T/4$ samples just to estimate $g(x^0)$. Then, you claimed that "In practice, sampling less may not significantly affect performance," which lacks evidence and support.

---

> ### Author Rebuttal · Authors · 2026-03-31
>
> Thanks for the positive feedback and appreciation of our work! Below, we clarify several points and highlight some noteworthy contributions of our work.
>
> ---
>
> **Q1.** The novelty is good but not very strong. The framework leans heavily on known ingredients: SGD bounds, UNIXGRAD, grid search, and prior ensemble ideas. The new part is mostly the self-bounding wrapper.
>
> **A1.** Thanks for the feedback! We believe the significance of this work lies in the **flexibility and simplicity of the self-bounding wrapper**, providing strong potential for wide applicability. Technically, this paper enhances the well-known grid-search framework with self-bounding and applies it to two classical settings, obtaining **competitive convergence with the SOTA methods using parameter-free methods**. Our framework uses UnixGrad and SGD only as the basic learner for the convex and non-convex cases, respectively. And these base learners can be replaced arbitrarily by any algorithms designed for convex or non-convex optimization, validating the flexibility of our framework.
>
> ---
>
> **Q2.** No experiments are provided. For a grid-search framework motivated partly by practice, this is a major omission. I am fully aware that this is a theory paper, yet, simultaneously, it proposes a new grid-search method. It would be good to see how it works in some small examples/algorithms/problems.
>
> **A2.** Thanks for the question. We have added experimental results to demonstrate the effectiveness of our method, with all results provided in [anonymous-link](https://anonymous.4open.science/r/ICML26-215Fd1/experiment.pdf).
> - **Setup.** We use ResNet-18 with an added linear layer (for convex case, only the linear layer is updated), CIFAR-10, cross-entropy loss, with total budget $T=10^4$ for initial sampling and training.
> - **Conclusions.** Our experiments conduct two key messages:
>   - Our method, *while being parameter-free*, obtains **competitive convergence compared with optimally tuned methods**.
>   - Reducing the initial sample budget has **little** impact, and may **improve** the results thanks to the saved budget.
>
> You can refer to the attachment or the response to Reviewer # dEJt for more details about the experiment setup and complete results.
>
> ---
>
> **Q3.** The practical oracle overhead is unclear. For example, Algorithm 2 spends $T/4$ samples just to estimate $g(x^0)$. Then, you claimed that "In practice, sampling less may not significantly affect performance," which lacks evidence and support.
>
> **A3.** Thanks for the question. We conduct experiments on multiple choices of the initial sampling budget and show that **reducing the initial sampling budget does not hurt the empirical performance**, and may even improve results by saving budget for the optimization phase. More experimental results are available at [anonymous-link](https://anonymous.4open.science/r/ICML26-215Fd1/experiment.pdf).
> This phenomenon can also be explained below: less initial sampling offers more budget for the later grid search, exhibiting a trade-off between initial exploration and later exploitation. Therefore, sampling less initially might not necessarily hurt the performance.
>
> ---
>
> We believe that our work has made valuable contributions and has potential to be extended to more settings.
> Once again, we appreciate your positive feedback. We would be happy to provide any further clarifications needed.

---

> > ### Author Rebuttal · Reviewer_1teK · 2026-04-04
> >
> > Thank you for the careful reply and explanation. I really appreciate the additional efforts in the experiments. I believe the work made a nontrivial, valuable contribution to the community. Hence, I maintain my positive evaluation. Good luck with the submission!

---

> > > ### Author Response · Authors · 2026-04-06
> > >
> > > Thank you again for recognizing the contribution of our work and for providing valuable suggestions to help improve it!

---

### Decision · Program_Chairs · 2026-04-30

**Decision:**

Accept (regular)

**Comment:**

This paper proposes GRASP, a grid-search framework with self-bounding analysis for fully parameter-free stochastic optimization. Reviewers agree that the paper is theoretically sound and conceptually elegant. Some concerns remain regarding practical efficiency (e.g., initial sampling cost and budget splitting), the lack of experiments, and clarity of assumptions. In particular, certain assumptions appear to apply primarily to the convex setting and should be stated more explicitly. After the rebuttal, most concerns were adequately addressed. Given the theoretical nature of this paper, I recommend acceptance.